# The cell surface hyaluronidase TMEM2 plays an essential role in mouse neural crest cell development and survival

Toshihiro Inubushi[1]*, Yuichiro Nakanishi[1], Makoto Abe[2], Yoshifumi Takahata[3], Riko Nishimura[3], Hiroshi Kurosaka[1], Fumitoshi Irie[4], Takashi Yamashiro[1], Yu Yamaguchi[4]

**1** Department of Orthodontics and Dentofacial Orthopedics, Osaka University Graduate School of Dentistry, Osaka, Japan, **2** Department of Oral Anatomy and Developmental Biology, Osaka University Graduate School of Dentistry, Osaka, Japan, **3** Department of Molecular and Cellular Biochemistry, Osaka University Graduate School of Dentistry, Osaka, Japan, **4** Human Genetics Program, Sanford Burnham Prebys Medical Discovery Institute, La Jolla, California, United States of America

* inubushi.toshihiro.dent@osaka-u.ac.jp

**Data Availability Statement:** All relevant data are within the manuscript and its Supporting Information files.

## Abstract

Hyaluronan (HA) is a major extracellular matrix component whose tissue levels are dynamically regulated during embryonic development. Although the synthesis of HA has been shown to exert a substantial influence on embryonic morphogenesis, the functional importance of the catabolic aspect of HA turnover is poorly understood. Here, we demonstrate that the transmembrane hyaluronidase TMEM2 plays an essential role in neural crest development and the morphogenesis of neural crest derivatives, as evidenced by the presence of severe craniofacial abnormalities in *Wnt1-Cre*–mediated *Tmem2* knockout (*Tmem2^{CKO}*) mice. Neural crest cells (NCCs) are a migratory population of cells that gives rise to diverse cell lineages, including the craniofacial complex, the peripheral nervous system, and part of the heart. Analysis of *Tmem2* expression during NCC formation and migration reveals that *Tmem2* is expressed at the site of NCC delamination and in emigrating Sox9-positive NCCs. In *Tmem2^{CKO}* embryos, the number of NCCs emigrating from the neural tube is greatly reduced. Furthermore, linage tracing reveals that the number of NCCs traversing the ventral migration pathway and the number of post-migratory neural crest derivatives are both significantly reduced in a *Tmem2^{CKO}* background. *In vitro* studies using *Tmem2*-depleted mouse O9-1 neural crest cells demonstrate that *Tmem2* expression is essential for the ability of these cells to form focal adhesions on and to migrate into HA-containing substrates. Additionally, we show that *Tmem2*-deficient NCCs exhibit increased apoptotic cell death in NCC-derived tissues, an observation that is corroborated by *in vitro* experiments using O9-1 cells. Collectively, our data demonstrate that TMEM2-mediated HA degradation plays an essential role in normal neural crest development. This study reveals the hitherto unrecognized functional importance of HA degradation in embryonic development and highlights the pivotal role of *Tmem2* in the developmental process.

**Funding:** This work was supported by grants-in-aid for scientific research programs from the Japan Society for the Promotion of Science (#19KK0232 to T.I., #20H03896 to T.I., #19KK0231 to T.Y.) and NIH R01 NS41332 and RF1 AG057579 (to Y.Y.). T. I. was the recipient of the Uehara Memorial Foundation Fellowship. The funders had no role in study design, data collection and analysis, decision to publish, or preparation of the manuscript.

**Competing interests:** The authors have declared that no competing interests exist.

## Author summary

As a major component of the extracellular matrix, hyaluronan is particularly abundant in the extracellular matrix of embryonic tissues, where its expression is dynamically regulated during tissue morphogenetic processes. Tissue levels of hyaluronan are regulated not only by its synthesis but also by its degradation. Curiously, however, mice lacking known hyaluronidase molecules, including HYAL1 and HYAL2, exhibit minimal embryonic phenotypes. As a result, our understanding of the role of the catabolic aspect of hyaluronan metabolism in embryonic development is quite limited. Here, we show that TMEM2, a recently identified hyaluronidase that degrades hyaluronan on the cell surface, plays a critical role in the development of neural crest cells and their derivatives. Our analyses of *Tmem2* conditional knockout mice, *Tmem2* knock-in reporter mice, and *in vitro* cell cultures demonstrate that TMEM2 is essential for generating a tissue environment needed for efficient migration of neural crest cells from the neural tube. Our paper reveals for the first time that the degradation of hyaluronan plays a specific regulatory role in embryonic morphogenesis, and that dysregulation of hyaluronan degradation leads to severe developmental defects.

## Introduction

Craniofacial anomalies, including midface hypoplasia and cleft lip and/or palate, account for one-third of all congenital birth defects [1]. Normal craniofacial development is an intricate biological process that requires the action of a number of distinct cell autonomous and cell non-autonomous factors and pathways. Among these, the extracellular matrix (ECM) plays a particularly important role. Neural crest cells (NCCs) are a migratory population of cells that arise at the edge of the neural tube during neurulation, and contribute to the formation of a variety of tissues, including the cardiovascular system, peripheral nervous system, skeleton, and craniofacial tissues [2]. After induction and specification at the edge of the neural tube, NCCs undergo a process called delamination, in which they emigrate from their site of origin and subsequently migrate toward target sites. Mutations in genes involved in NCC development can lead to a wide range of human congenital malformations, including craniofacial anomalies.

Hyaluronan (HA) is a non-sulfated glycosaminoglycan widely distributed in the ECM of a variety of tissues. The importance of HA in the development of NCC-derived tissues is illustrated by the observation that disruption of HA synthesis by means of genetic ablation of the *Has* genes, which encode hyaluronan synthases, leads to defects in NCC-derived tissues. In mice, *Has2* ablation results in mid-gestation (E9.5–10) lethality due to severe defects in endocardial cushion formation [3,4]. Defects in these cardiac structures are often associated with the abnormal development of a subpopulation of cranial NCCs [5]. Cranial NCC-targeted conditional *Has2* knockout mice also exhibit these defects in formation of the NCC-derived craniofacial structures [6]. In *Xenopus*, mutations in hyaluronan synthase genes (*has1* and *has2*) result in impaired NCC migration and craniofacial defects [7]. It has also been shown that the HA-binding proteoglycans aggrecan and versican exerts inhibitory effects on NCCs migration [8,9].

Tissue levels of ECM molecules are controlled by the balance between synthesis and degradation. A unique feature of HA among ECM molecules is its extremely rapid turnover [10–12]. An estimated one-third of the total body HA (~15 g in a person with a 70 kg body weight) is degraded and turned over daily [11], and the half-life of HA in skin is only 1 to 1.5 days [10].

Although the rate of HA degradation in embryonic tissues has not been determined, it seems reasonable to speculate that HA is actively turned over during this period, since HA staining changes dynamically in intensity during embryonic development [13–15]. Although several classes of hyaluronidase proteins have been identified, the identity of a cell surface (or secretory) hyaluronidase(s) that is directly involved in the dynamic remodeling of the extracellular HA matrix has remained elusive. The HYAL family proteins, including HYAL1, HYAL2, and SPAM1, have been extensively studied as principal hyaluronidases in mammalian species. However, while some data suggest that a subpopulation of HYAL proteins is present on the cell surface [16,17], their physiological sites of action have been controversial. In fact, multiple lines of evidence indicate that HYAL proteins are primarily associated with lysosomes and related intracellular vesicles [18–23]. Also, HYAL1 and HYAL2 favor acidic pH (pH < 5) for their hyaluronidase activity [21,24], consistent with the possibility that they are primarily involved in the degradation of internalized HA fragments in lysosomes. Moreover, mice with mutations in HYAL proteins exhibit only mild developmental phenotypes. *Hyal1* null mice are viable and exhibit few developmental defects [25]. *Hyal2* null mice on a C57Bl/6 background are also viable and exhibit only mild developmental abnormalities, including shortening of the nose and widened interorbital space. On a mixed C129;CD1;C57Bl/6 background, approximately half of adult *Hyal2* null mice exhibit moderate heart defects, including expanded heart valves and cardiac hypertrophy [26,27]. These relatively mild developmental phenotypes resulting from knockout of HYAL family hyaluronidases suggest that there may be another hyaluronidase responsible for the prominent role of HA degradation during embryonic development.

Transmembrane protein 2 (TMEM2; gene symbol *CEMIP2*) was originally identified as a large type II transmembrane protein with an unknown function. Even in the absence of a defined function, zebrafish *tmem2* mutants (*frozen ventricles* and *wickham*) were nevertheless found to exhibit a developmental heart phenotype related to endocardial cushion defect, accompanied by excessive accumulation of HA [28,29]. Subsequently, we demonstrated that TMEM2 is a hyaluronidase that degrades extracellular high-molecular weight HA into fragments as small as ~5 kDa at near neutral pH [30]. In mouse embryos, *Tmem2* is expressed in a developmentally regulated manner, with the peak of expression prior to E11 and with prominent sites of expression in the neural tube, the first branchial arch, and the frontonasal processes [31]. More recently, we have demonstrated that TMEM2 plays a critical role in promoting integrin-mediated cell adhesion and migration via its removal of anti-adhesive HA from focal adhesion sites [32]. These observations suggest to us that TMEM2 may be the key hyaluronidase that regulates dynamic remodeling of the HA-rich ECM during embryonic development.

In this report, we use *Wnt1-Cre*–mediated conditional ablation of *Tmem2* to determine the role of this cell surface hyaluronidase in NCC development and embryonic morphogenesis. We show that ablation of *Tmem2* leads to severe craniofacial defects, attesting to the essential role of TMEM2-mediated HA degradation in NCC development and survival. Our results provide novel insight into the critical importance of dynamic HA turnover and matrix remodeling in embryonic development.

## Results

### *Wnt1-Cre* mediated conditional *Tmem2* knockout results in defects in craniofacial development

To gain initial insight into the role of TMEM2 during NCC development, we used *in situ* hybridization to examine the spatiotemporal expression pattern of *Tmem2* during the mid-

gestation period. In whole mount preparations at E8.5 and 9.0, robust *Tmem2* expression is detected in the neural tube, frontonasal region, branchial arches, and heart (S1A Fig). Transverse sections of the E9.0 neural tube further demonstrate that *Tmem2* expression in the neural tube is concentrated mainly in its dorsal region (S1A Fig, *Transverse*), from which NCCs arise. At E9.5 and E10.5, robust *Tmem2* expression is also seen in tissues with contributions from NCCs, including the forebrain, midbrain, hindbrain, trigeminal ganglion, branchial arches, heart, and dorsal root ganglia (S1B Fig).

The *Wnt1-Cre* driver has been used to determine the function of genes in NCC development, migration, and subsequent differentiation [33,34]. To determine the role of *Tmem2* in the development of NCCs and their derivatives, we crossed a conditional *Tmem2* allele (*Tmem2$^{flox}$*) into *Wnt1-Cre* mice (see Materials and Method). While heterozygous *Wnt1-Cre; Tmem2$^{flox/wt}$* mice were born alive without detectable developmental defects and were fertile, no homozygous conditional mutants (i.e., *Wnt1-Cre;Tmem2$^{flox/flox}$*; hereafter referred to as *Tmem2$^{CKO}$*, hereafter) were recovered at birth. Therefore, we performed timed matings to obtain homozygous embryos at several time points between E10.5 and E12.5. At E10.5, *Tmem2$^{CKO}$* embryos exhibit hypoplasia of the frontonasal, maxillary, and mandibular processes (Fig 1A, *open arrowheads*). Hemorrhage and edema were frequently observed in the craniofacial regions of *Tmem2$^{CKO}$* embryos at times later than E10.5 (Fig 1A, *arrow*). In addition, growth retardation is often observed in *Tmem2$^{CKO}$* embryos at this stage. At gestational stages later than E12.5, 100% (42 of 42 embryos) of *Tmem2$^{CKO}$* embryos exhibit severe craniofacial abnormalities, characterized by reduced outgrowth of the frontonasal and maxillary processes, lack of fusion of the medial nasal and mandible processes at the midline, lack of fusion between frontonasal process and maxillary process, and wide nasal cavity (Fig 1B and 1C). Histomorphological analysis of E12.5 *Tmem2$^{CKO}$* embryos further demonstrates the hypoplastic, laterally expanded maxillary components in these mice and the absence of fusion of the facial processes (Fig 1D-ii and 1D-iv). Epithelial blistering was frequently observed in the lateral portion of the frontonasal process and in the midline region of the mandibular arches (34 of 42 embryos, 81.0%) (*arrows* in Fig 1D-ii and 1D-iv). Defects in branchial arch derivatives, such as the tongue, are also observed in *Tmem2$^{CKO}$* embryos. NCC-derived peripheral nervous tissues, such as trigeminal, facial, and vestibular ganglia, are consistently smaller in *Tmem2$^{CKO}$* embryos than in control embryos (Fig 1D-vi). In addition, neural tube defects, including exencephaly, were detected in a fraction of *Tmem2$^{CKO}$* embryos (4 of 42 embryos, 9.5%) (Figs 1B and S2). No live *Tmem2$^{CKO}$* embryos were recovered past E13.5. All *Tmem2$^{CKO}$* embryos recovered at E13.5 exhibited severe craniofacial and cardiovascular abnormalities. Cardiac NCCs, a subpopulation of cranial NCCs, migrate into the third, fourth and sixth branchial arches and give rise to the aortic and pulmonary trunk, the cap of the intraventricular septum, the developing outflow tract cushions, and the parasympathetic system of the heart [35]. This is consistent with the fact that *Wnt1-Cre* is also active in cardiac NCCs [36]. Consistent with the functional role for TMEM2 in cardiac NCCs, *Tmem2$^{CKO}$* embryos exhibit expanded endocardial cushions and lack of the aorticopulmonary septum in the outflow tract region (S3 Fig).

To determine whether TMEM2 functions as a hyaluronidase in tissues that exhibit morphogenetic defects in *Tmem2$^{CKO}$* mice, we examined whether HA is increased in these tissues. HA staining using biotinylated HA binding protein (bHABP) reveals increased HA in the frontonasal process of E12.5 of *Tmem2$^{CKO}$* embryos (Fig 1E). Similar increases in HA staining are observed in the mandible and in the outflow tract endocardial cushions of the heart (S4 Fig). Increased tissue HA levels are confirmed by quantification of HA in lysates of craniofacial tissues (Fig 1E). Moreover, hematoxylin and eosin (H&E) staining of the frontonasal process reveals that the size of extracellular space is significantly increased in *Tmem2$^{CKO}$* embryos (Fig 1F), a tissue phenotype generally associated with excess HA accumulation [37]. Altogether,

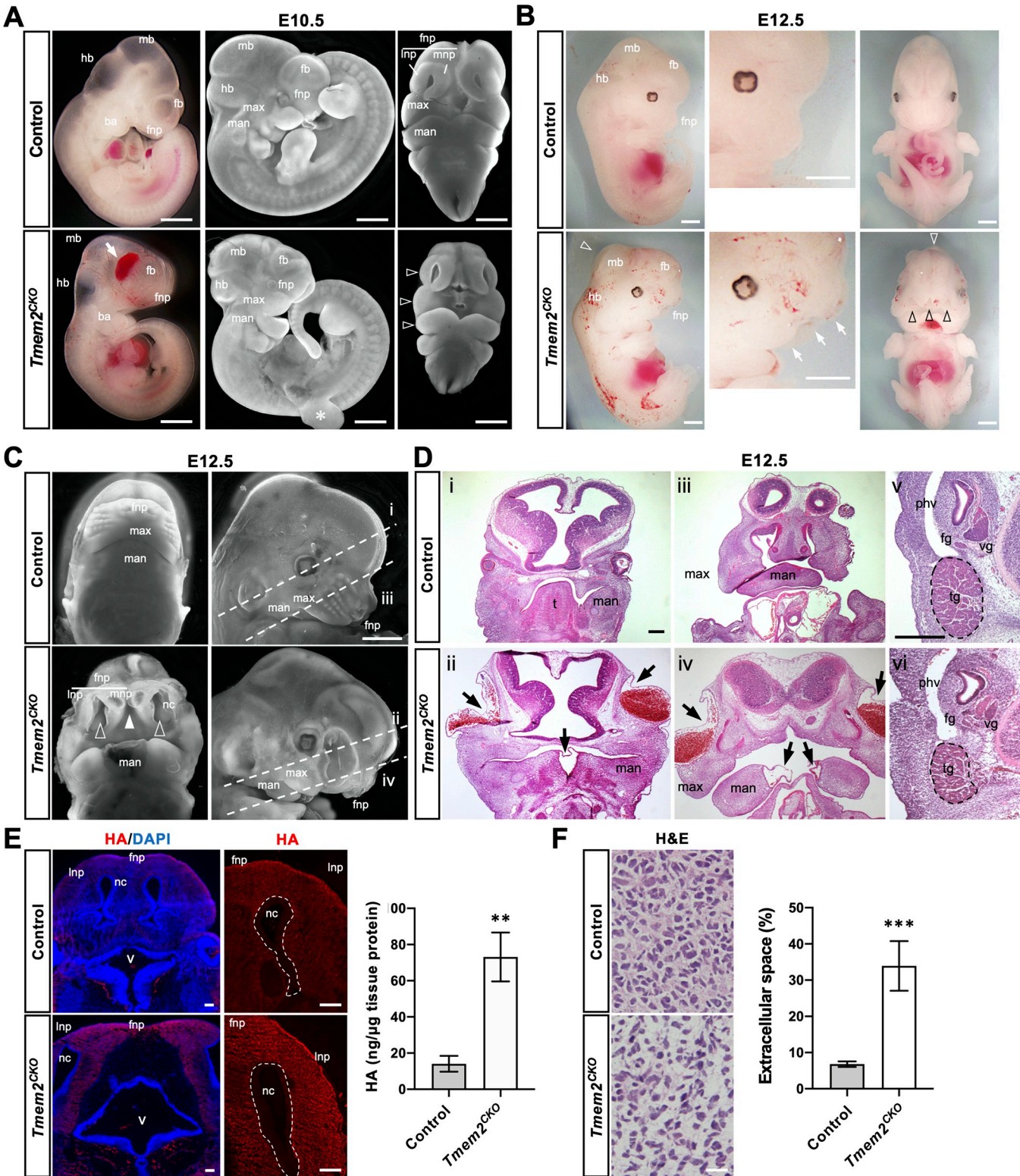

**Fig 1. NCC-targeted conditional knockout of *Tmem2* leads to craniofacial malformations.** (**A**) Gross phenotype of *Tmem2^CKO* embryos at E10.5. Whole-mount images of unstained embryos (*left panels*) and fluorescence microscopic images of DAPI-stained embryos (*center* and *right panels*) are shown. Hypoplasia of the frontonasal process, maxillary process, and mandibular process is observed in *Tmem2^CKO* (*open arrowheads*). *Arrow* indicates edema in the facial region. *Asterisk* indicates a partial tear of the right forelimb bud inadvertently caused during dissection. (**B**) Gross phenotype of *Tmem2^CKO* embryos at E12.5. Whole-mount images reveal severe developmental malformations of the craniofacial region of *Tmem2^CKO* embryos, with marked hypoplasia of the facial

structures (*white arrows*) and lack of fusion of the frontonasal processes (*black open arrowheads*; see Fig 1C for more detailed images). *White open arrowhead* indicates mild exencephaly observed in some of the $Tmem2^{CKO}$ embryos. Hemorrhagic lesions are also frequently observed in $Tmem2^{CKO}$ embryos. (**C, D**) Phenotypes in the craniofacial region of E12.5 $Tmem2^{CKO}$ embryos. (**C**) Fluorescence microscopic images of the craniofacial region of DAPI-stained embryos. $Tmem2^{CKO}$ embryos show lack of fusion of the bilateral medial nasal processes (*filled arrowhead*) and bilateral mandible processes at the midline, lack of fusion between frontonasal process and maxillary process (*open arrowheads*), and wide opening of the nasal cavity. *Broken lines* (i, ii, iii and iv) indicate the orientations of sections shown in *D*. (**D**) H&E staining of transverse sections through the forebrain (i, ii) and the maxillary region (iii, iv). (v, vi) High magnification images corresponding to the area of NCC-derived peripheral nervous tissues, including trigeminal (*tg*), vestibular (*vg*), and facial (*fg*) ganglia. Note that the size of these NCC-derived ganglia is reduced in $Tmem2^{CKO}$ embryos. $Tmem2^{CKO}$ embryos also exhibit blister-like epithelial detachment, often accompanied with hemorrhaging, in the lateral portion of the frontonasal process and at the midline of the mandibular arches (*arrows*). (**E**) Accumulation of HA in craniofacial tissues of E12.5 $Tmem2^{CKO}$ embryos. Transverse sections of the facial processes were double-labeled with bHABP and/or DAPI. Increased HA staining is observed in the facial processes of $Tmem2^{CKO}$ embryos (*HA*). *Bar graph* shows the quantification of HA in the facial tissue of E12.5 embryos (ng/μg tissue protein). Data represent means ± SD ($n = 5$). $**p < 0.01$ by Student's *t*-test. (**F**) H&E staining of the frontonasal tissue demonstrates expanded extracellular space in E12.5 $Tmem2^{CKO}$ and control embryos. *Bar graph* shows the quantification of the size of the extracellular space in the frontonasal tissue. Data represent means ± SD ($n = 5$). $***p < 0.001$ by Student's *t*-test. *ba*, branchial arch; *fnp*, frontonasal process; *max*, maxillary process; *man*, mandibular process; *fb*, forebrain; *mb*, midbrain; *hb*, hindbrain; *nc*, nasal cavity; *mnp*, medial nasal process; *lnp*, lateral nasal process; *t.* tongue; *phv*, primary head vein; *tg*, trigeminal ganglion; *v*, fourth ventricle; *vg*, vestibular ganglion; *fg*, facial ganglion. Scale bars, 500 μm in **A**-**D**; 250 μm in **E**; 25 μm in **F**.

these observations on $Tmem2^{CKO}$ embryos demonstrate that $Tmem2$ is required for normal cranial neural crest development via its action as a functional hyaluronidase.

## Expression of TMEM2 in developing tissues relative to the distribution of HA

To gain insight into cellular mechanisms underlying the phenotype of $Tmem2^{CKO}$ embryos, we compared the detailed spatiotemporal expression of TMEM2 protein relative to the distribution of HA in NCCs and their derivatives. To facilitate this analysis, we created a knock-in mouse line (referred to as $Tmem2\text{-}FLAG^{KI}$ mice, hereafter), in which a FLAG-tagged TMEM2 is expressed under the control of the endogenous $Tmem2$ promoter (see Materials and Methods and S5 Fig). Mice homozygous for this knock-in allele are born and grow normally, indicating the functionality of TMEM2 produced from this FLAG-tagged allele.

Analysis of $Tmem2\text{-}FLAG^{KI}$ embryos at E11.0 reveals that TMEM2 is strongly expressed in the neuroepithelia of the forebrain, midbrain, and hindbrain, the facial prominence, the branchial arches, the dorsal root ganglia, and the developing heart (Figs 2A and 2B and S6). These results are in good agreement with the expression of $Tmem2$ mRNA detected by *in situ* hybridization (see S1 Fig) and confirm that TMEM2 protein is expressed in tissues populated by NCC derivatives. Overall, the sites of TMEM2 expression coincide closely with the sites where defects are observed in $Tmem2^{CKO}$ embryos. Moreover, double-staining for TMEM2 and HA using anti-FLAG antibody and bHABP, respectively, reveals that TMEM2 protein and HA exhibit roughly inverse patterns of distribution (Fig 2A and 2B), supporting the validity of a role of TMEM2 as a functional hyaluronidase in these tissues. Together, these observations are consistent with the notion that TMEM2 acts as a functional hyaluronidase in the context of the development of NCC-derived tissues.

## Role of TMEM2 in NCC emigration

Based on the *in situ* hybridization data showing distinct expression of $Tmem2$ mRNA in the dorsal neural tube (see S1 Fig), we used $Tmem2\text{-}FLAG^{KI}$ mice to examine the expression pattern of TMEM2 protein during neural tube closure and neural crest formation. In E9.0 embryos, TMEM2 is expressed in the neural fold and the neural plate border, while HA staining is absent in these regions (Fig 2C). Next, we analyzed the expression of TMEM2 in NCCs by double-labeling with Sox9. In mice, Sox9 is the first Group E Sox gene to be expressed within the neural crest and serves as a marker for pre-migratory NCCs [38]. As shown in Fig 2D, Sox9-positive cells in the dorsal neural tube and those that have emigrated out of the

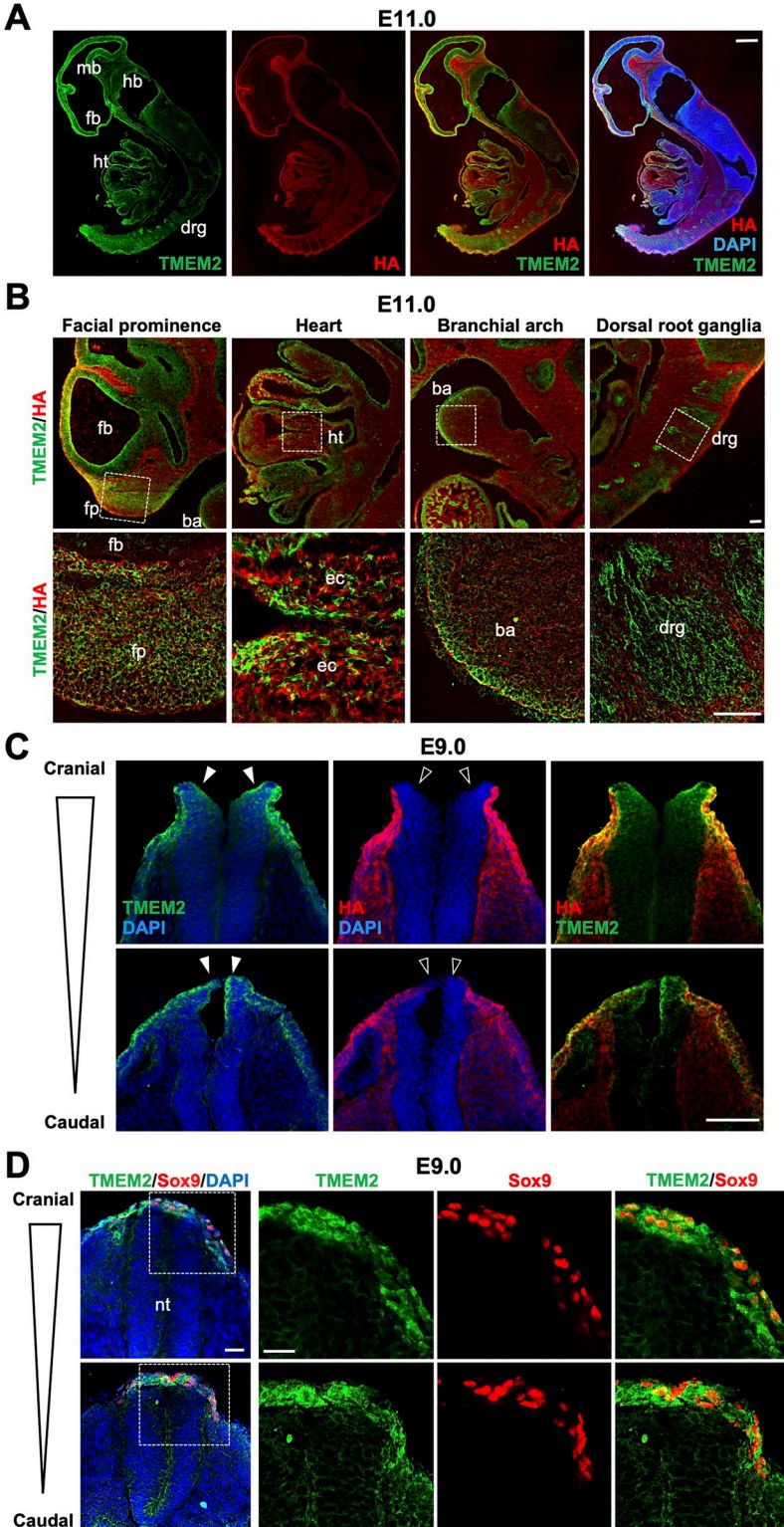

**Fig 2. Expression of TMEM2 relative to HA distribution in the mouse embryo.** (**A**) Sagittal sections of *Tmem2-FLAG^KI* reporter embryos at E11.0 were double-labeled with anti-FLAG antibody and bHABP. Nuclei were counterstained with DAPI. *fb*, forebrain; *mb*, midbrain; *hb*, hindbrain; *ht*, heart; *drg*, dorsal root ganglia. (**B**) High magnification images of the facial prominence, heart, branchial arch, and dorsal root ganglion in *Tmem2-FLAG^KI* embryos at E11.0 double-labeled for TMEM2-FLAG protein and HA. Areas indicated by boxes are enlarged in lower

panels. *fb*, forebrain; *fp*, facial prominence; *ba*, branchial arch; *ht*, heart; *ec*, endocardial cushion; *drg*, dorsal root ganglia. (**C**) Transverse sections of the neural tube of *Tmem2-FLAG^KI* embryos at E9.0. Sections at the cranial and trunk levels of the neural tube were double-labeled for TMEM2-FLAG protein and HA as in *A*. TMEM2 expression is observed in the neural plate and the border region of the neural tube (*filled arrowheads*), whereas these sites are devoid of HA staining (*open arrowheads*). (**D**) Double-labeling of neural crest cells for TMEM2-FLAG and Sox9. Transverse sections of E9.0 neural tube were stained for TMEM2-FLAG and Sox9. Sox9-positive pre-migratory and emigrating NCCs at the edge of the neural tube highly express TMEM2. *nt*, neural tube. Scale bars, 250 μm in **A**; 50 μm in **B**; 300 μm in **C**; 100 μm in **D**.

neural tube highly express TMEM2 protein, suggesting that TMEM2 may play a functional role in NCC emigration from the neural tube.

To define the functional role of TMEM2 in NCC emigration, we examined the localization of NCCs in *Tmem2^CKO* and control embryos using Sox9 and Sox10 as markers for pre-migratory and migratory NCCs, respectively (Fig 3). Sox10 expression is initiated in NCCs as they dissociate from the neural tube and persists during NCC migration [39]. In wild-type control embryos at E9.0, Sox9 expression was seen in NCCs in the neural tube and in the emigrating NCCs (Fig 3A, *Sox9/Control*). Significantly, the region occupied by these cells was devoid of HA staining (Fig 3A, *open arrowheads* in *HA/Control*), suggestive of TMEM2-dependent removal of HA in the pericellular space of pre-migratory NCCs. In contrast, in *Tmem2^CKO* embryos, few Sox9-expressing cells were observed outside of the neural tube (Fig 3A, *Sox9/ Tmem2^CKO*) and HA staining persists in the dorsal midline region of the neural tube (**Fig 3A**, *filled arrowheads* in *HA/Tmem2^CKO*). Sox10 staining (**Fig 3B**) indicates that the number of Sox10-expressing cells in the ventral migration pathway is reduced in *Tmem2^CKO* embryos (Fig 3B, *Tmem2^CKO*). Quantification of NCCs within the neural tube and NCCs along the ventral migration pathway (Fig 3C) confirms these observations and further suggests impaired emigration of *Tmem2*-deficient NCCs. On the other hand, neither the epithelial-mesenchymal transition nor the specification of NCCs are altered in the absence of TMEM2 (S7 Fig). Together, these results suggest that functional TMEM2 is required for the efficient emigration of NCCs out of the neural tube.

## Lineage tracing of NCCs in *Tmem2^CKO* mice

We next performed lineage tracing experiments to determine the contribution of *Tmem2*-deficient NCCs to the formation of NCC-derived structures. For this, we crossed the ZsGreen reporter gene onto the *Tmem2^CKO* background to generate *Tmem2^CKO;ZsGreen* mice. These mice express ZsGreen upon Cre-mediated removal of a floxed-stop-flox cassette [40]. Distribution patterns of ZsGreen-positive NCCs were compared between these and control (*Wnt1-Cre;Tmem2^wt/wt;ZsGreen*) mice. While the gross distribution pattern of ZsGreen-positive neural crest derivatives is not significantly altered, both the spatial extent and intensity of ZsGreen signals in the facial prominences and the branchial arches are reduced at E9.5 in the *Tmem2^CKO* background compared to the control background (Fig 4A, *open arrowheads*). In addition, ZsGreen-positive cells aberrantly accumulate in the dorsal region of the midbrain and hindbrain in *Tmem2^CKO;ZsGreen* embryos (Fig 4A, *filled arrowheads*). *Tmem2^CKO; ZsGreen* embryos also exhibit an abnormally folded midbrain neuroepithelium that is ZsGreen-positive (Fig 4B, *bracket*), which may also be a consequence of aberrant migration of *Tmem2*-deficient NCCs from the dorsal midbrain. Sections through the frontonasal process and the midbrain of E10.5 embryos reveal that ZsGreen signals in the trigeminal ganglia are significantly reduced in *Tmem2^CKO;ZsGreen* embryos (Fig 4B, *arrowheads*), suggesting that NCC contribution to these structures is reduced. Sox10 immunostaining confirms that the contribution of post-migratory NCCs to these structures is indeed reduced in *Tmem2^CKO;*

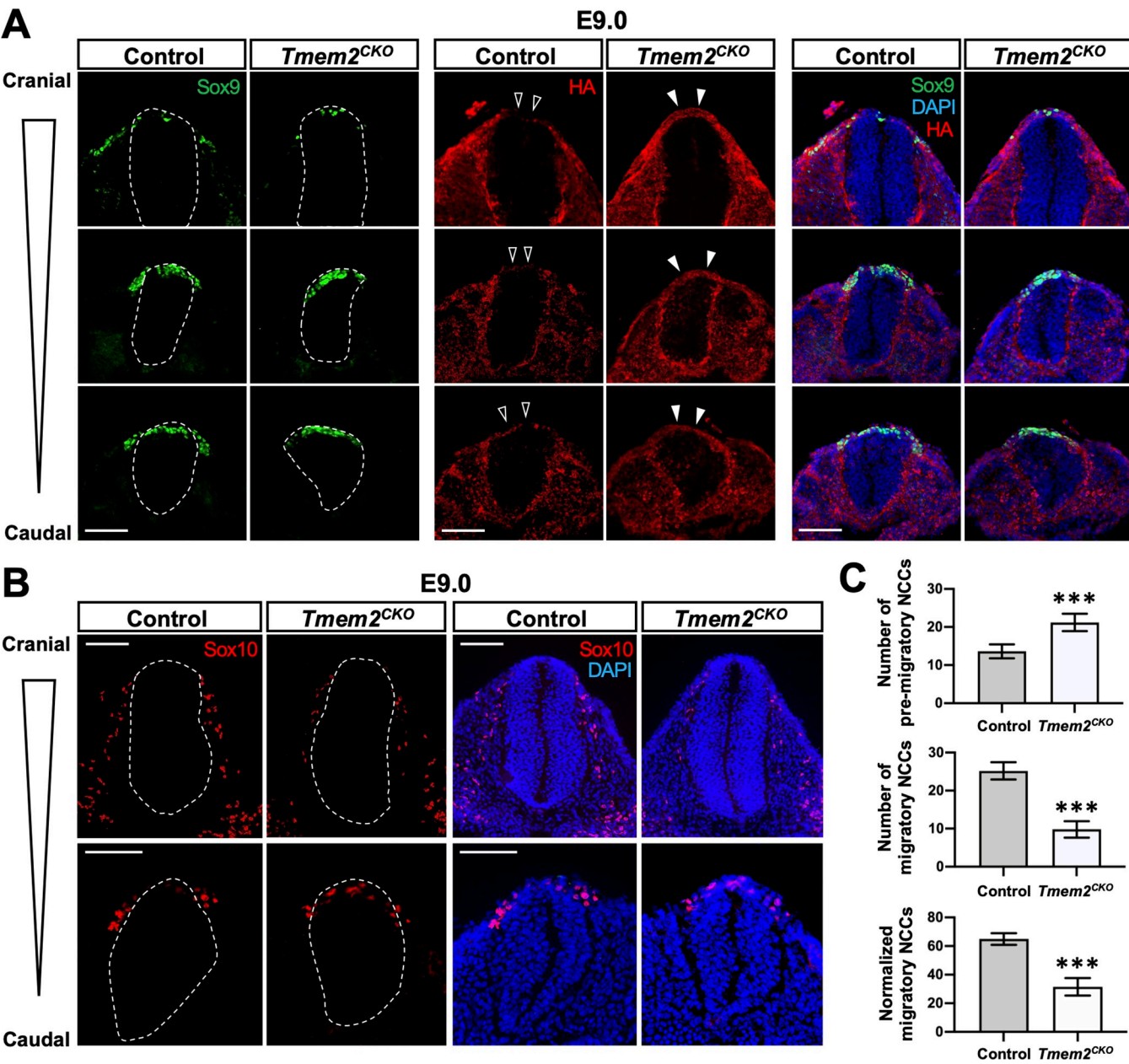

**Fig 3. Effects of *Tmem2* ablation on NCC development.** (**A**) Transverse sections at three different levels of the neural tube of *Tmem2*^*CKO*^ and control embryos at E9.0 were triple-stained for Sox9 (*green*), HA (*red*), and nuclei (*blue*). *Broken lines* indicate the external contour of the neural tube. Note that emigration of Sox9-positive cells out of the neural tube is reduced in *Tmem2*^*CKO*^ embryos. Also, in *Tmem2*^*CKO*^ embryos, the dorsal surface of the neural tube exhibits HA staining (*filled arrowheads*), whereas little HA staining is detectable in the corresponding area of control embryos (*open arrowheads*). (**B**) Transverse sections of *Tmem2*^*CKO*^ and control embryos at E9.0 were double-stained for Sox10 (*red*) and nuclei (*blue*). In *Tmem2*^*CKO*^ embryos, the number of Sox10-positive cells outside of the neural tube is significantly decreased. The image in the caudal neural tube reveals that some Sox10-positive cells persist within the neural tube in *Tmem2*^*CKO*^ embryos. (**C**) Quantification of pre-migratory and migrating NCCs in *Tmem2*^*CKO*^ and control embryos. The numbers of Sox9-positive cells within the neural tube and Sox10-positive cells outside of the neural tube were counted as pre-migratory and migrating NCCs, respectively (*top* and *middle bar graphs*). The ratio of migratory NCCs relative to the total number of NCCs was also compared between *Tmem2*^*CKO*^ and control embryos (*bottom graph*). Data represent means ± SD (*n* = 5). ***$p < 0.001$ by Student's *t*-test. Scale bars, 300 μm in **A** and **B**.

*ZsGreen* embryos (Fig 4B, *middle panels*). In *Tmem2*^*CKO*^*;ZsGreen* embryos, the contribution of ZsGreen-positive cells to trunk neural tube-derived dorsal root ganglia is reduced (**Fig 4C,** *arrowheads*), whereas ZsGreen-positive cells within the neural tube are increased in number

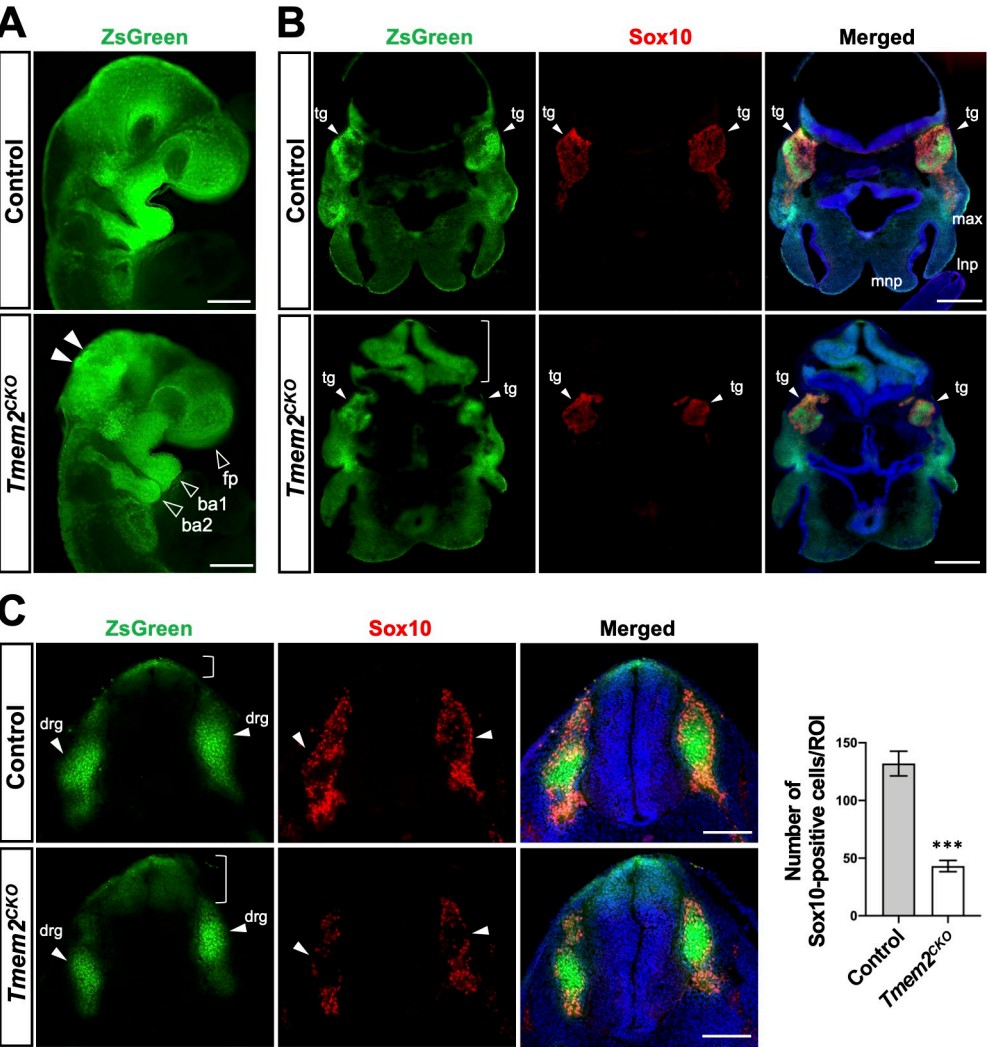

**Fig 4. Lineage tracing of NCCs. (A)** Whole-mount fluorescence images of *Tmem2^CKO^;ZsGreen* and control embryos at E9.5. In *Wnt1-Tmem2^CKO^;ZsGreen* embryos, domains occupied by ZsGreen-labeled NCCs are smaller and the intensity of ZsGreen signals is reduced in the facial prominence and in the first and second branchial arches (*open arrowheads*). Aberrant accumulation of *ZsGreen*-positive cells is observed in the dorsal midbrain and hindbrain of *Tmem2^CKO^;ZsGreen* embryos (*filled arrowheads*). **(B)** Transverse sections of *Tmem2^CKO^;ZsGreen* and control embryos at E10.5 stained for Sox10. *Arrowheads* indicated trigeminal ganglia (*tg*). The number of Sox10-positive NCCs arrived at the trigeminal ganglia is significantly reduced in *Tmem2^CKO^;ZsGreen* embryos. *Bracket* indicates abnormal folding of the hindbrain neuroepithelium. **(C)** Transverse sections through the caudal neural tube of *Tmem2^CKO^;ZsGreen* and control embryos at E10.5 stained for Sox10. In *Tmem2^CKO^;ZsGreen* embryos, the number of NCCs migrated to the dorsal root ganglia (*arrowheads*) is decreased, while the number remaining in the neural tube is increased (*brackets*). *Bar graph* shows the quantification of Sox10-positive cells in the dorsal root ganglia at E10.5. Data represent mean ± SD ($n = 5$). ***$p < 0.001$ by unpaired Student's *t*-test. Scale bars, 500 μm in **A-C**. *fp*, facial prominence; *ba1*, the first branchial arch; *ba2*, the second branchial arch; *lnp*, lateral nasal process; *max*, maxillary process; *mnp*, medial nasal process; *tg*, trigeminal ganglion; *drg*, dorsal root ganglia.

and their distribution is spatially expanded ventrally (Fig 4C, *bracket*). Impaired contribution of *Tmem2*-deficient NCCs to dorsal root ganglia was also demonstrated by immunostaining for Sox10 and counting of Sox10-positive cells (Fig 4C, *Bar graph*). On the other hand, overall dorsoventral patterning of the neural tube is not altered in *Tmem2^CKO^* embryos (S8 Fig). Together, these results support the notion that *Tmem2* deficiency perturbs specific aspects of

NCC emigration from the neural tube, resulting in reduced contribution of these cells to NCC-derived structures.

## *Tmem2*-deficient O9-1 mouse neural crest cells are defective in migration in HA-rich environments

A considerable amount of evidence indicates that the migration of NCCs is mediated by integrins and the formation of focal adhesions (FAs) [41–45]. While a thin HA coating on glass or plastic can mediate cell attachment, thick gel-like HA present in the pericellular and extracellular matrices interferes with the engagement of adhesion receptors with their ECM ligands, thus forming a repulsive barrier to cell adhesion and migration [37,46–49]. The migratory route of NCCs, including the dorsolateral aspect of the neural tube into which early NCCs emigrate, contains high levels of HA [14] (see also Fig 2C). Thus, it is conceivable that NCCs use TMEM2 to degrade this HA barrier as a means of allowing integrin-ECM engagement and FA formation. In this scenario, the NCC phenotype of *Tmem2*$^{CKO}$ mice would stem from the inability of *Tmem2*-deficient NCCs to establish robust integrin-mediated interactions with the HA-rich ECM surrounding the neural tube.

To address this hypothesis, we generated *Tmem2*-depleted O9-1 mouse neural crest cells via lentivirus-mediated delivery of shRNA [50] (see S9 and S10 Figs for the characterization of these cells). Using these and control transfected O9-1 cells, we examined the effect of *Tmem2* depletion on their adhesive and migratory behavior on HA-containing substrates. In a cell-based assay of hyaluronidase activity [30], control O9-1 cells exhibit a potent ability to degrade HA (Fig 5A). Detailed analysis of the HA degradation pattern reveals that control O9-1 cells degrade substrate-bound HA in a pattern that resembles the distribution of FAs (Fig 5B), as previously observed in a variety of cell lines [32]. Immunostaining for the FA marker vinculin confirms that control O9-1 cells form FAs at the sites of HA degradation (Fig 5B). *Tmem2*-depleted O9-1 cells, in contrast, exhibit markedly reduced HA degradation (Fig 5A). Moreover, vinculin immunostaining reveals that FA formation on HA-containing substrates is greatly attenuated in *Tmem2*-depleted cells (Fig 5B). Thus, O9-1 cells employ TMEM2 to mediate HA degradation in the ECM as a means of promoting FA formation.

To model NCC emigration into HA-rich tissues surrounding the neural tube, we performed a wound healing-type migration assay on mixed substrates consisting of type I collagen (Col1) and high molecular weight (HMW) HA species (average molecular weight 1200–1400 kDa). This *in vitro* model allows us to mimic the emigration of leading edge NCCs into the HA-rich interstitial ECM. Control O9-1 cells are highly migratory on these mixed substrates; cells at the boundary of the wound readily enter into the HA-rich gap (Fig 5C, *Control shRNA*). In contrast, entry into the gap is significantly reduced in *Tmem2*-depleted O9-1 cells (Fig 5C, *Tmem2 shRNA*). Significantly, *Tmem2* depletion does not affect the entry of O9-1 cells into mixed substrates consisting of Col1 and low molecular weight (an average molecular weight 4–6 kDa) HA species (S11 Fig). Since TMEM2 is not capable of cleaving HA species smaller than ~5 kDa [30], this result is consistent with the notion that the impaired migration of *Tmem2*-depleted cells on Col1/HMW HA substrates is due to the absence of TMEM2-mediated degradation of HMW-HA. Together, these results demonstrate the functional importance of TMEM2 in the migration of O9-1 cells in an HA-rich environment.

To examine the *in vivo* relevance of these findings, we analyzed the localization of vinculin in NCCs of *Tmem2*$^{CKO}$*;ZsGreen* and control (i.e., *Wnt1-Cre;Tmem2*$^{wt/wt}$*;ZsGreen*) embryos at E9.5 (Fig 5D). In control embryos (Fig 5D, *Control*), individual NCCs emigrating from the neural tube exhibit marked cortical vinculin immunoreactivity, lining the periphery of individual cells (*insets*). A similar cortical vinculin accumulation has previously been reported for

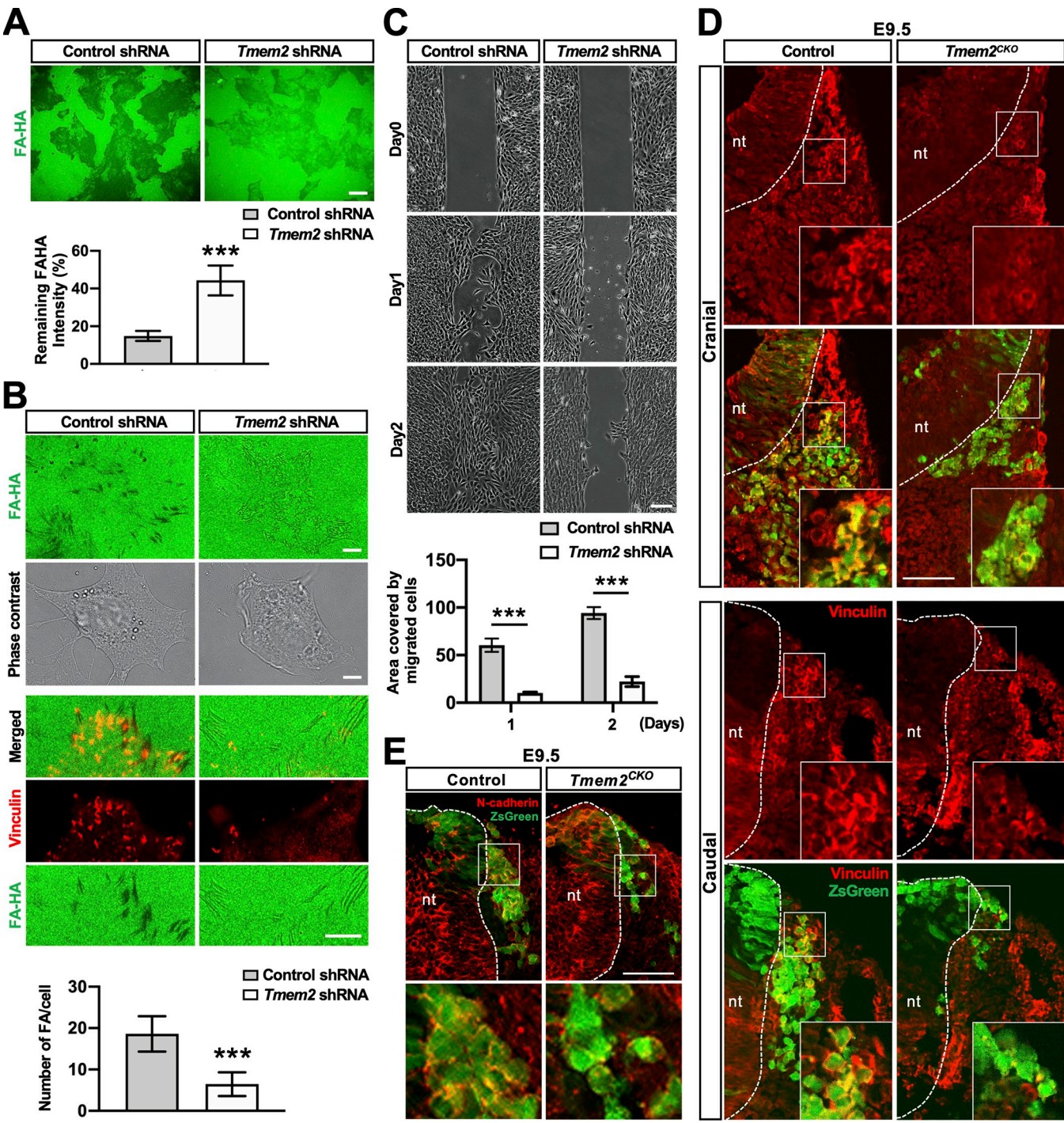

**Fig 5. *Tmem2*-depleted NCCs are defective in FA formation and migration in HA-rich environments both *in vitro* and *in vivo*.** (**A**) Cell-based hyaluronidase assay. *Tmem2*-depleted and control O9-1 cells were cultured for 48 h on glass coverslips coated with fluoresceinated HA (FA-HA). HA degrading activity is revealed as dark areas in the fluorescent background. The level of HA degradation was also quantitatively compared between *Tmem2*-depleted and control O9-1 cells as described in Materials and Methods (*bar graph*). Data represent mean ± SD of the fluorescence intensity underneath a cell relative to that in cell-free area (n > 50 cells per condition pooled from three independent experiments). ***$p < 0.001$ by unpaired Student's *t*-test. (**B**) O9-1 cells degrade substrate-bound HA at FAs. Cell-based hyaluronidase assays were performed for 16 h and cells were stained for vinculin. In control O9-1 cells, HA degradation occurs coincident with vinculin-positive FAs. In *Tmem2*-depleted O9-1 cells, HA degradation and FA formation are greatly diminished. The number of FAs per cell was quantitatively compared between *Tmem2*-depleted and control O9-1 cells (*bar graph*). Data represent mean ± SD (n >30 cells per condition pooled from three independent experiments). ***$p < 0.001$ by unpaired Student's *t*-test. (**C**) Representative images of the migration of *Tmem2*-depleted and control O9-1 cells into a cell-free gap on Col1/HA mixed substrates. *Top panels* show images of gaps immediately after removal of the ibidi 2-well

Culture-Insert. Other panels show images of gaps after a 24 h or 48 h incubation. Data are representative of three independent experiments. *Bar graph* shows the quantitative analysis of cell migration. Data represent the mean ± SD of the gap area covered by migratory cells relative to the area of the original gap (n = 5 per condition). ***$p < 0.001$ by two-way ANOVA with Bonferroni's multiple comparison test. (**D, E**) Transverse sections of the neural tube in the caudal and cranial regions of E9.5 *Tmem2^{CKO};ZsGreen* and control embryos were stained for vinculin (**D**) or N-cadherin (**E**). (**D**) Vinculin accumulation in the cellular cortex is reduced in NCCs in *Tmem2^{CKO};ZsGreen* embryos. Insets show enlarged images of migrating NCCs in the boxed areas. (**E**) Cell surface expression of N-cadherin is reduced in NCCs in *Tmem2^{CKO};ZsGreen* embryos. *Lower panels* show enlarge images of the boxed areas. *nt*, neural tube. Scale bars, 25 μm in **A**; 2.5 μm in **B**; 200 μm in **C**; 150 μm in **D** and **E**.

cranial NCCs in wild-type mouse embryos [51]. In contrast, overall vinculin immunoreactivity in *Wnt1-Tmem2^{CKO};ZsGreen* embryos is greatly diminished in emigrating NCCs (Fig 5D, *Tmem2^{CKO}*), and little cortical vinculin localization is observed in these cells (*insets*). In addition, cell surface association of N-cadherin is reduced in NCCs of *Wnt1-Tmem2^{CKO};ZsGreen* embryos (Fig 5E). Overall, these results strongly suggest that the reduced NCCs emigration observed in *Tmem2^{CKO}* mice is caused primarily by the impaired ability of NCCs to overcome the barrier effect of HA and form FAs.

### *Tmem2* deficiency induces apoptosis in the neural crest-derived tissues

It has been shown that genetic ablation of certain types of integrins or their cognate adhesion partners leads not only to reduced NCC migration but also to increased NCC apoptosis [52]. Given the markedly reduced outgrowth of facial processes in *Tmem2^{CKO}* embryos (see Fig 1), we asked whether *Tmem2* deletion affects the proliferation and/or survival of NCCs in craniofacial tissues. TUNEL assays reveals that the number of TUNEL-positive cells is significantly increased in the lateral and medial nasal processes and branchial arches of E12.5 *Tmem2^{CKO}* embryos compared with that seen in control embryos (Figs 6A and S12A). Increased cell death is rather specific for facial processes and branchial arches, and TUNEL-positive cells are not increased in the developing heart of *Tmem2^{CKO}* embryos (S12 Fig). Interestingly, the number of TUNEL-positive cells observed during the process of migration is also not altered by *Tmem2* deficiency, either (S13A Fig). Moreover, there is little difference in the number of TUNEL/ZsGreen double-positive cells between mutant and control embryos in E9.0 craniofacial tissues (S13B Fig). Together, these results suggest that apoptosis of *Tmem2*-deficient NCCs occurs after their arrival in the craniofacial tissue. In contrast to cell death, cell proliferation does not appear to be affected by *Tmem2* ablation. Immunostaining for phospho-histone H3 (PHH3) in these NCC-derived tissues demonstrates no detectable difference in the number of proliferative cells between *Tmem2^{CKO}* and control embryos (Figs 6B and S12B).

To corroborate these findings *in vitro*, we quantitatively analyzed the incidence of apoptosis in O9-1 cells on HA containing and HA deficient substrates. For this, *Tmem2*-depleted and control O9-1 cells were cultured for 48 h on Col1 alone or on Col1/HA mixed substrates [32]. Apoptotic cell death was analyzed by double-staining with anti-annexin V antibody and propidium iodide (PI), followed by flow cytometric analysis of live, early apoptotic, and later apoptotic cells [53]. On pure Col1 substrates, there is no difference in the number and pattern of apoptotic (annexin V/PI double-positive) cells between *Tmem2*-depleted and control O9-1 cells (Fig 6C). On the other hand, on mixed Col1/HA substrates, a significantly greater proportion of *Tmem2*-depleted O9-1 cells undergo apoptosis than observed in control O9-1 cells (Fig 6D). Thus, these *in vitro* data demonstrate that, while the absence of TMEM2 has no bearing on cell survival on HA deficient substrates, the expression of TMEM2 is important for promoting the survival of cells in HA-rich environments. This support the notion that the apoptotic phenotype of NCC-derived craniofacial components in *Tmem2^{CKO}* embryos is a direct consequence of the loss of TMEM2 function.

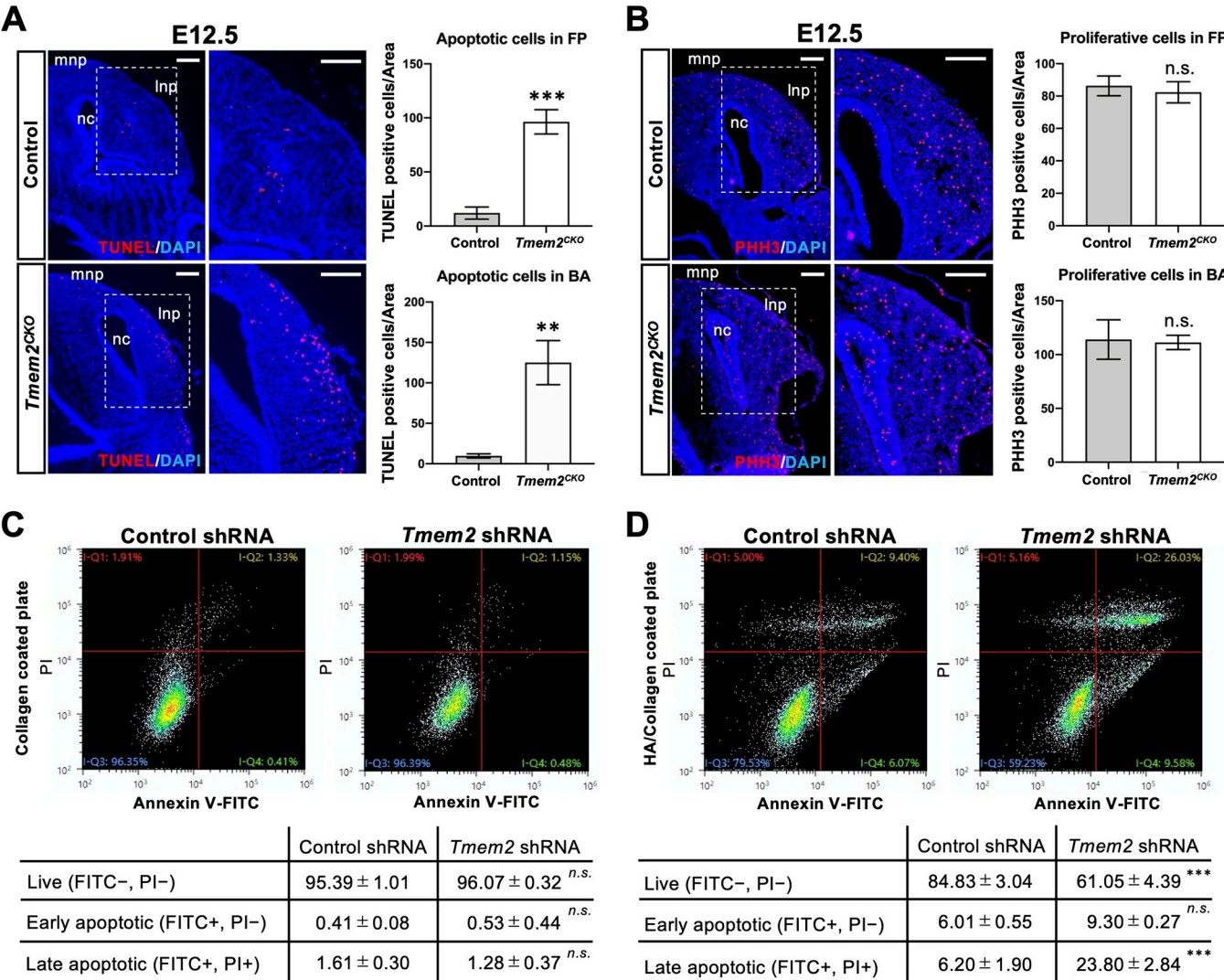

**Fig 6. Increased apoptosis of post-migratory NCCs in *Tmem2*<sup>CKO</sup> embryos.** (A) Analysis of cell death in the facial processes of E12.5 embryos. Transverse sections through the facial processes were analyzed by TUNEL assays. *Bar graphs* on the right represent the number of TUNEL-positive cells per unit area of the facial processes (*top*) and the first branchial arch (*bottom*). Means ± SD ($n = 3$) are shown as horizontal bars. **$p < 0.01$, ***$p < 0.001$ by unpaired Student's *t*-test. (B) Analysis of cell proliferation in the facial processes of E12.5 embryos. Transverse sections through the facial processes were stained with anti-PHH3 antibody. *Bar graphs* on the right represent the number of PHH3-positive cells per unit area of the facial processes (*top*) and the first branchial arch (*bottom*). *mnp*, medial process; *lnp*, lateral nasal process; *nc*, nasal cavity. Means ± SD ($n = 3$) are shown as horizontal bars. *n.s.*, not significant by unpaired Student's *t*-test. (C, D) Analysis of apoptosis of *Tmem2*-depleted and control O9-1 cells cultured on HA-free and HA-containing substrates. *Tmem2*-depleted and control O9-1 cells were cultured for 48 h on Col1 alone (C) and on Col1/HA mixed (D) substrates. Cells were then stained with propidium iodide (PI) and anti-annexin V antibody, followed by the analysis of apoptosis by flow cytometry. *Tables* show the quantification of live (lower left quadrant in the charts), early apoptotic (lower right quadrant), and late apoptotic (upper right quadrant) cells. ***$p < 0.001$; *n.s.*, not significant by unpaired Student's *t*-test. Scale bars, 250 μm in A and B.

## Discussion

Subsequent to the molecular identification of HA synthase genes, studies using mutant animals in which these genes are mutated or deleted have provided genetic evidence for the functional importance of HA in embryonic development [3,6,7]. On the other hand, despite the fact that tissue levels of HA are dynamically regulated by extremely rapid HA turnover, much less is known about the role of HA degradation in embryonic development. The paucity of data related to this issue is largely due to uncertainty regarding the identity of the

hyaluronidase(s) responsible for extracellular HA degradation in developing embryos. In the present study, we demonstrate a critical role for the cell surface hyaluronidase TMEM2 in neural crest development, based on our findings from a series of *in vivo* and *in vitro* studies. In wild-type embryos, *Tmem2* is expressed in locations associated with NCC generation and differentiation, including the midline of the neural tube, the branchial arches, the endocardium, and the craniofacial primordia. Consistent with the notion that TMEM2 functions as a hyaluronidase, these regions are generally devoid of HA in wild-type embryos, whereas the same areas in *Tmem2^{CKO}* mice exhibit high levels of HA accumulation. NCC-targeted ablation of *Tmem2* results in defective morphogenesis of NCC derivatives, most notably the craniofacial structure, confirming the essential role of this cell surface hyaluronidase in neural crest development.

Several lines of evidence support the conclusion that these *Tmem2*-dependent defects in NCC derivatives are primarily due to impaired emigration of *Tmem2*-deficient NCCs. Lineage tracing analysis of *Tmem2^{CKO}* embryos reveals accumulation of Sox9-positive pre-migratory cells in the neural tube, while the number of Sox10-positive migrating NCCs is significantly reduced. In addition, *in vitro* experiments using O9-1 mouse neural crest cells demonstrate that *Tmem2* expression is required for the ability of NCCs to assemble robust focal adhesions and to efficiently migrate into HA-containing substrates. In contrast, loss of *Tmem2* expression does not have significant effects on the patterning of the neural tube or the specification of NCCs (S8 Fig). Taken together, these results indicate that TMEM2-mediated HA degradation is essential for efficient NCC migration from the neural tube and thus plays an important functional role in the development of NCC derivatives.

We have recently reported that tumor cells use TMEM2 to remove matrix-associated HA in the vicinity of adhesion sites, thereby facilitating the formation of mature FAs and promoting cell migration. This action of TMEM2 is coordinated with integrin function via direct interactions between the two proteins [32]. These tumor-related observations have direct mechanistic relevance to normal neural crest development, because the functional importance of integrins and FAs in NCC emigration and migration has been demonstrated by numerous studies [41–45,54,55]. In fact, it has recently been shown that inhibition of any of the key FA components, namely integrin-β1, vinculin, or talin, leads to impairment of the onset of NCC migration *in vivo* [41]. We propose that the critical contribution of TMEM2 to NCC migration is its ability to remove ECM-associated HA that would otherwise sterically interfere with the ability of integrins to engage with ECM ligands. In other words, without TMEM2, leading NCCs may be incapable of achieving the integrin-ECM engagement that is necessary for robust FA assembly. Consistent with this model, *Tmem2*-depleted O9-1 cells exhibit reduced entry into HA-containing gaps in *in vitro* wound healing assays, and furthermore, they are not capable of efficiently forming FAs on these substrates (Fig 5B and 5C). We also demonstrate that *Tmem2*-ablated NCCs *in vivo* are incapable of recruiting vinculin to subcortical areas (Fig 5D), supporting the *in vivo* relevance of our model. Taken together, our results strongly suggest that TMEM2 plays a functional role in NCC emigration via the creation of a microenvironment that is permissive for integrin-ECM engagement. On the other hand, our present data do not provide a definitive answer regarding a possible requirement for TMEM2 in the subsequent migration of NCCs that have emigrated from the neural tube. This process is governed primarily by a collective migration mechanism [56,57], which does not typically rely on the formation of FAs by individual cells. Still, it is possible that the effect of TMEM2 on cell adhesion may not be limited to integrin-dependent cell-ECM adhesions. This possibility is suggested by the observation of reduced cell surface association of N-cadherin in *Wnt1-Tmem2^{CKO};ZsGreen* embryos (Fig 5E). The possible role for TMEM2 in N-cadherin–mediated cell-cell adhesion is

thus an issue that needs to be addressed in order to fully understand the function of this novel hyaluronidase in NCC migration.

It has been shown that short HA fragments and oligomers are biologically active and can modulate cell behavior, including motility and migration [58,59]. TMEM2 has been suggested to play a role in the generation of such biologically active HA fragments [60,61], although no direct evidence supporting such a function of TMEM2 is currently available. Therefore, while our current data demonstrate the critical importance of TMEM2 for cell-ECM adhesion, it cannot be ruled out that TMEM2 may also be involved in the regulation of NCC behavior via generation of putative motility-stimulating HA fragments. Another remaining issue is to understand how the synthesis and degradation of HA is orchestrated during NCC development [62]. Inactivation of HA synthesis has been reported to result in craniofacial abnormalities [6,7]. It is curious why both the removal and accumulation of HA appear to cause superficially similar craniofacial phenotypes. While addressing this issue would require further studies combining mutations of hyaluronan synthase genes and TMEM2, it is tempting to speculate that the level of HA needs to be tightly regulated within a narrow range for normal development of NCCs and their derivatives. Studies on integrin-mediated cell migration have demonstrated this type of behavior—i.e., both excessive and insufficient adhesion impede cell migration [63,64]. HA may exert similar effects on NCC migration.

In addition to our observations on NCC migration, another notable finding of our current study is the increased apoptosis of post-migratory NCCs in craniofacial tissues of $Tmem2^{CKO}$ mice (Fig 6A). This $in vivo$ finding is further supported by results from $in vitro$ experiments with $Tmem2$-depleted O9-1 cells (Fig 6D). There are several potential mechanisms, not necessarily mutually exclusive, by which TMEM2 could be functionally involved in post-migratory NCC survival. Given the ability of TMEM2 to facilitate FA formation and cell adhesion in HA-rich environments [32], it is possible that $Tmem2$-deficiency renders NCCs susceptible to anoikis, a form of programmed cell death that is induced in anchorage-dependent cells by loss of adhesion [65]. Without TMEM2, NCCs may not be able to establish the robust cell adhesion needed to avoid anoikis in an HA-rich environment. Another potential mechanism for the increased apoptosis of post-migratory $Tmem2$-deficient NCCs may be elevated endoplasmic reticulum (ER) stress in the absence of TMEM2—In searching for genes needed for cells to survive ER-based stress due to protein misfolding, Schinzel et al. [61] serendipitously identified TMEM2 as a potent protective factor against ER stress damage. In any case, the determination of the precise mechanism by which TMEM2 protects NCCs from apoptosis requires further investigation.

In addition to abnormalities in craniofacial tissues, $Tmem2^{CKO}$ embryos also exhibit cardiovascular abnormalities. This observation is consistent with previous reports on zebrafish $tmem2$ mutants [28,29]. In this vein, it is interesting to note that mice with mutations in genes involved in HA metabolism and function, including $Has2$ (encoding the HA synthase Has2) and $Vcan$ (encoding the HA-binding proteoglycan versican), also exhibit cardiovascular abnormalities [3,66,67], illustrating the critical importance of HA in the development of the heart and great vessels. Future studies using additional Cre deleter lines to target vascular lineages will help address the specific role of TMEM2 in cardiovascular development.

In conclusion, our results demonstrate that TMEM2 plays a critical role in neural crest development. In this regard, our paper reveals for the first time that the catabolic aspect of HA metabolism has a specific regulatory role in embryonic morphogenesis, and that dysregulation of this mechanism leads to severe developmental defects. It is therefore noteworthy that a non-synonymous 1358G>A (C453Y) mutation in exon 5 of the $TMEM2$ gene, which is predicted to be pathogenic, has been identified in an individual with hypertelorism, myopia, retinal dystrophy, abnormality of the sternum, joint laxity, and inguinal hernia (ClinVar, NCBI, NIH:

accession number: VCV000827846 and VCV000827847). It remains to be seen whether mutations/polymorphisms in *TMEM2* are associated with specific craniofacial malformation in humans.

## Materials and methods

### Ethics statement

All animal experiments were performed in accordance with the guidelines of the Animal Care and Use Committee of the Osaka University Graduate School of Dentistry, Osaka, Japan. The Committee on the Ethics of Animal Experiments of the same university approved the study protocol. Approval number: 04263, 29-024-0.

### Mice

A conditional *Tmem2* null allele (*Tmem2$^{flox}$*) was created by Cyagen Biosciences (Santa Clara, CA) using TurboKnockout gene targeting methods. Mouse genomic fragments containing homology arms and the conditional knockout region were amplified from a BAC library and were sequentially assembled into a targeting vector together with recombination sites and selection markers. The targeting vector was electroporated into C57BL/6-derived ES cells, followed by drug selection and isolation of drug-resistant clones. The resultant *Tmem2$^{flox}$* allele contains two loxP sites flanking exons 4 and 5 of the *Tmem2* gene, so that these two exons will be deleted by Cre-mediated recombination. Exon 5 harbors amino acid residues (R265, D273, D286), mutagenesis of any of which abrogates the hyaluronidase activity of TMEM2 [30]. Moreover, conjugation of exon 3 to exon 6 due to the deletion of exons 4 and 5 results in a frameshift and a premature stop codon. Mice carrying homozygous *Tmem2$^{flox}$* alleles are developmentally normal and fertile, confirming that the non-recombined *Tmem2$^{flox}$* allele is fully functional. Mice with conditional *Tmem2* ablation targeted to NCCs were generated by crossing the *Tmem2$^{flox}$* allele and the *Wnt1-Cre* transgene [36]. *Wnt1-Cre* mice in a C57BL/6 background were obtained from Prof. Sachiko Iseki of Tokyo Medical and Dental University. Resultant *Wnt1-Cre;Tmem2$^{flox/wt}$* male mice were mated with *Tmem2$^{flox/flox}$* or *Tmem2$^{flox/wt}$* female mice to obtain *Tmem2* conditional knockout mice (i.e., *Wnt1-Cre;Tmem2$^{flox/flox}$*). Littermates inheriting an incomplete combination of the above alleles were used as controls. Genotyping of mice and embryos was performed by PCR with the specific primers listed in **S1 Table**, using DNA prepared from tail biopsies and yolk sacs. For *in vivo* fate mapping experiments, ZsGreen reporter mice (R26;ZsGreen; JAX mice 007906) were purchased from The Jackson Laboratory (Bar Harbor, ME). Mice with the *Tmem2$^{flox}$* allele and the *R26;ZsGreen* transgene were crossed to create *Tmem2$^{flox/wt}$;ZsGreen* mice. *Wnt1-Cre;Tmem2$^{flox/wt}$* male mice were mated with *Tmem2$^{flox/wt}$;ZsGreen* female mice to obtain ZsGreen reporter mice with *Tmem2* conditional knockout (*Tmem2$^{flox/flox}$;Wnt1-Cre;ZsGreen*) or control allele (*Tmem2$^{wt/wt}$;Wnt1-Cre;ZsGreen*). Primer sequences for genotyping are listed in **S1 Table.**

### Creation of the *Tmem2*-FLAG knock-in allele

Pronuclear stage embryos from C57BL6/J mice were purchased from ARK Resource (Kumamoto, Japan). Recombinant Cas9 protein, crRNA and tracrRNA were obtained from Integrated DNA technology. Single-stranded oligodeoxynucleotides for insertion of a FLAG epitope were designed with 30 bp sequence homologies on each side of the Cas9-mediated double strand break (see S2 Table for sequence information). For generation of *Tmem2*-FLAG knock-in (*Tmem2-FLAG$^{KI}$*) mice, we used the Technique for Animal Knockout System by Electroporation (TAKE) [68]. Briefly, embryos were washed twice with Opti-MEM solution

and aligned in the electrode gap filled with 50 μl of Cas9/gRNA(crRNA-tracrRNA complex)/ ssODN (200/100/100 ng/μl) mixture. The intact embryos were subjected to electroporation using poring (225V) and transfer pulses (20V). After electroporation, embryos were returned to KSOM Mouse Embryo Media (Millipore Sigma) at 37°C. Genome edited 2-cell embryos were transferred to oviducts of pseudopregnant ICR female mice, and genomic DNA from newborn mice was analyzed by PCR (see S1 Table for primer sequences).

## Histology

Embryos were fixed in 4% paraformaldehyde (PFA) in PBS, decalcified in EDTA, embedded in paraffin, and sectioned at 5 μm thickness. For frozen sections, embryos were fixed in 4% PFA, equilibrated in graded sucrose, and embedded in Tissue-Tek (OCT compound, Sakura). Hematoxylin and eosin (H&E) staining was performed for the assessment of cellular and tissue structure. Quantification of the relative size of the extracellular space was performed according to Senf et al. [69]. Briefly, images were captured from H&E-stained sections, and cellular areas were outlined according to the H&E staining intensity. The outlined cellular area was calculated using Image J 1.51s. The extracellular space was calculated by subtracting the cellular area from total area.

## Immunohistochemistry, TUNEL staining, and *in situ* hybridization

Embryos were fixed with 4% PFA and incubated at 4°C overnight with **DAPI** (4′,6-diamidino-2-phenylindole) diluted 1**:1000 (Dojindo, #D523-10)**. Immunostaining of frozen sections was performed as previously described [70]. The following antibodies were used in this study: mouse monoclonal anti-FLAG (F9291, 1:200), rabbit polyclonal anti-FLAG (F7425, 1:200), mouse monoclonal anti-vinculin (V4505, 1:200), and mouse monoclonal α-SMA (A5228, 1:200) from Sigma; mouse anti-N-cadherin, anti-Nkx2.2, anti-Pax7 and anti-Pax3 from Developmental Studies Hybridoma Bank (DSHB); rabbit polyclonal anti-Sox9 (#82630, 1:100) from Cell Signaling Technology; rabbit polyclonal anti-Sox10 (ab27655, 1:100) from abcam; mouse monoclonal anti-phospho-histone H3 (PHH3) (05-746R, 1:200) from MilliporeSigma; Alexa 488-labelled goat anti-mouse IgG (A28175, 1:500), Alexa 488-labelled goat anti–rabbit IgG (#A11034, 1:500), Alexa 555-labelled goat anti-mouse IgG (A21422, 1:500) and Alexa 555-labeled goat anti-rabbit IgG (A21428, 1:500), Alexa 555-labeled goat anti-rat IgG (A21434, 1:500), Alexa 555-labeled streptavidin (S32355, 1:500) from Invitrogen; DAPI were purchased from Invitrogen. Apoptotic cells were identified by using an *in situ* cell death detection kit (Roche, catalog #11684795910) according to the manufacturer´s instructions. Whole mount nuclear fluorescence imaging was utilized to capture morphological features of embryonic tissue. Biotinylated HA binding protein (bHABP) was used for detection of HA as previously described [71]. Whole-mount *in situ* hybridization was performed as described previously [72]. The digoxigenin-labelled antisense RNA probes were produced using a DIG RNA labeling kit (Roche, #11277073910) according to the manufacturer's protocol. A minimum of three embryos of each specimen type were examined per probe.

## Immunoprecipitation and immunoblotting

Immunoprecipitation was performed with Dynabeads streptavidin magnetic beads and biotinylated anti-FLAG antibody (MilliporeSigma, #F9291), according to the manufacturer's instructions. Briefly, for detecting endogenous C-terminal FLAG-tagged TMEM2, cell lysates were obtained by treating embryos with lysis buffer containing 1% NP-40, 25 mM HEPES (pH 7.5), 150 mM NaCl, and 5 mM $MgCl_2$. Wild-type embryos were used as a negative control for the experiment. Dynabeads streptavidin magnetic beads (Invitrogen, DB65801) were washed 3

times with wash buffer and incubated with the biotinylated anti-FLAG antibody in PBS for 30 min at room temperature using gentle rotation. The antibody-coated streptavidin magnetic beads were separated with a magnet and the coated beads were washed 4–5 times in PBS containing 0.1% BSA. Washed antibody-coated streptavidin magnetic beads were added to the cell lysates and incubated overnight at 4˚C. Antibody-coated streptavidin magnetic beads were captured with a magnet and washed 2–3 times. Immunoprecipitated materials were solubilized and subjected to SDS-PAGE on an 8–16% Tris-glycine gel (ThermoFisher), followed by electroblotting onto an Immobilon PVDF membrane (EMD Millipore). Membranes were incubated with rabbit polyclonal antibody against TMEM2 (Sigma Aldrich, SAB2106587) and then horseradish peroxidase-conjugated goat anti-rabbit IgG (Invitrogen, #1706565). The ECL Western Blotting Substrate (Nacalai Tesque, #07880) was used for detection.

## Quantification of HA

Neural crest-derived craniofacial tissues were dissected from $Tmem2^{CKO}$ and control embryos at E12.5. Following incubation in ice-cold lysis buffer for 30 min as described above, samples were centrifuged at 15,000 rpm for 20 min at 4˚C to prepare tissue lysates. Tissue HA contents were measured by latex-sensitized immunoturbidimetry (Hyaluronic Acid LT Assay, Fujifilm Wako Chemicals GmbH, Neuss, Germany) using a Hitachi 917s.

## Culture and lentiviral transduction of O9-1 cells

The **O9-1** mouse cranial neural crest cell line was purchased from MilliporeSigma (SCC049) and cultured in complete ES cell medium containing 15% FBS and LIF (MilliporeSigma, ES-101-B) with 100 units/ml penicillin-streptomycin (Invitrogen) at 37˚C in a humidified atmosphere containing 5% $CO_2$ according to the manufacturer's instructions. For osteogenic or chondrogenic differentiation, O9-1 cells were cultured with StemMACS OsteoDiff Media or StemMACS ChondroDiff Media (Milteny Biotec), respectively. Osteogenic differentiation and chondrogenic differentiation were evaluated by alizarin red or alcian blue staining as previously described [70,73]. To knockdown $Tmem2$ expression in O9-1 cells, we used lentivirus-mediated shRNA transduction. Lentivirus particles expressing an shRNA that is validated to deplete mouse $Tmem2$ (Mission shRNA, TRC Clone ID: TRCN0000295501, MilliporeSigma) and control lentivirus particles expressing an shRNA that does not target any known genes (Mission shRNA, MilliporeSigma SHC005) were purchased from MilliporeSigma. Lentivirus particles were added to O9-1 cells cultured in growth media supplemented with 5 μg/ml polybrene and cultured for 2 days. Cells transduced with lentiviral shRNAs were selected and maintained in the presence of 10 μg/mL puromycin.

## RNA extraction and qPCR analysis

Total RNA was isolated using RNeasy mini kit (Qiagen) and reverse transcribed to cDNA using an oligo (dT) with reverse transcriptase (Takara Bio), as previously described [74]. For real-time PCR, cDNA were amplified with the Fast SYBR Green Master Mix (Applied Biosystems, Foster City, CA) or TaqMan Fast Universal PCR Master Mix (Applied Biosystems, Foster City, CA). Data acquisition and analysis were performed with a Step One Real-Time PCR System using Step One Software, Version 2.1 (Applied Biosystems). The PCR products were quantified using $Gapdh$ as the reference gene. The TaqMan probes and primer sets for detection of $Tmem2$ (Mm00459599_m1) and $Gapdh$ (Mm99999915_g1) were purchased from Applied Biosystems. Primers for detection of $Itgb1$, $Itgb3$, $Itga1$ and $Itga2$ were shown in S3 Table.

### *In situ* HA degradation assay

This assay was performed as described previously [32]. Briefly, glass coverslips were coated with a mix of type I collagen and fluorescein-labeled HMW HA (Iwai Chemicals, Tokyo, Japan). Trypsinized cells were seeded onto the coated coverslips in 24-well plates at a density of $5 \times 10^4$ cells per coverslip and then incubated for 16 h or 48 h at 37°C in a $CO_2$ incubator. Cells were fixed with 4% paraformaldehyde in PBS overnight at 4°. In some experiments, fixed cells were permeabilized with PBS containing 0.2% Triton X-100 for 10 min and blocked with PBS containing 1% IgG-free bovine serum albumin (BSA) (Sigma, A2058) for 30 min at RT, followed by immunocytochemistry with mouse monoclonal anti-vinculin antibody (1:200 dilution in 1% BSA–PBS) (Sigma; V9264, clone hVIN-1) and Alexa 555–labelled goat anti–mouse IgG (#A21422, 1:500 dilution in 1% BSA–PBS). Stained coverslips were mounted in fluorescence mounting medium (Dako), and fluorescent images were captured using an all-in-one fluorescence microscope (BZ-X700, Keyence, Osaka, Japan). Quantification of the degraded HA area was performed by calculating fluorescence intensity in cellular areas using Image J 1.51s. The cellular area itself was also measured using Image J by outlining the cell area in phase contrast images that were taken simultaneously with fluorescence images of the same region. Total fluorescence intensity in the area underneath each cell was measured, which was then divided by the cellular area to derive the fluorescence intensity per area. The hyaluronidase activity was expressed as a ratio relative to the fluorescence intensity in the cell-free area (i.e., no degradation of HA).

### Cell migration assay

A wound healing-type assay using defined 500 μm cell-free gaps in Col1/HA substrates was performed utilizing 2-well culture inserts (ibidi; 80209). Glass coverslips were coated with type I collagen and unlabeled HA samples of various sizes, as described above. After drying, 2-well culture inserts (ibidi; 80209) were attached to coated coverslips. Insert-attached coverslips were then transferred into 24-well plates, and the inserts were filled externally with PBS. Cells ($1 \times 10^4$ cells in 70 μl of culture medium per insert) were seeded into the wells and cultured for 2 days. Two days later, culture inserts were detached from coverslips, and coverslips were transferred into new 24-well plates with fresh culture medium. At 24 h and 48 h in culture, phase contrast images were captured by an all-in-one fluorescence microscope (BZ-X700, Keyence, Osaka, Japan). The area of the 500 μm gap covered by migrating cells was analyzed by ImageJ 1.51s (NIH).

### Image analysis of NCC migration

For quantification of pre-migratory and migrating NCCs, transverse sections of the neural tube at corresponding truncal levels were prepared from E9.0 *Tmem2*$^{CKO}$ and control embryos. For identifying pre-migratory NCCs, sections were stained with anti-Sox9 antibody and DAPI, as described above. The numbers of Sox9-positive cells within the neural tube and Sox10-positive cells outside of the neural tube were counted as pre-migratory and migratory NCCs, respectively. For identifying NCCs that migrated to DRGs, sections were stained with anti-Sox10 antibody and DAPI, and a square area (1200 × 1200 μm in Fig 3C; 2000 × 2000 μm in Fig 4C) was selected as an ROI. In both analyses, the number of labeled cells in each ROI is counted at least 5 consecutive sections per embryo in a total 3–5 embryos by ImageJ 1.51s. The ratios of pre-migratory and migratory cells relative to total NCCs were also calculated.

### Statistical analysis

Statistical methods were not used to predetermine sample size. Statistical analyses were performed with GraphPad Prism 8. Two-sided Student's *t*-test and two-way ANOVA were used

under the assumption of normal distribution and observance of similar variance. A $p$ value of <0.05 was considered significant. Bonferroni post hoc analysis was performed where applicable. Values are expressed as mean ± SD. Data shown are representative images; each analysis was performed on at least three mice per genotype. Immunostaining was performed at least in triplicate.

## Supporting information

**S1 Fig. Expression of *Tmem2* in wild-type embryos analyzed by *in situ* hybridization.** (**A**) Whole-mount images of the dorsal and lateral aspects of embryos at E8.5 and E9.0 and a transverse section through the neural tube at E9.0. Robust *Tmem2* expression is observed in the dorsal midline region of the neural tube (*arrowheads*), the facial prominence, the branchial arches, and the heart. (**B**) Whole-mount *in situ* hybridization images at E9.5 and E10.5. *ba*, branchial arch; *drg*, dorsal root ganglia; *fb*, forebrain; *fp*, facial prominence; *h*, heart; *hb*, hindbrain; *mb*, midbrain; *tg*, trigeminal ganglia. Scale bars, 250 μm in **A**; 500 μm in **B**. (TIFF)

**S2 Fig. *Tmem2^CKO^* embryos exhibit abnormalities in the neural tube.** Images show dorsal views of *Tmem2^CKO^* and control embryos at E12.5. A fraction (4 of 42, 9.5%) of *Tmem2^CKO^* embryos exhibit defects in the neural tube, including incomplete neural tube closure (*filled arrowheads*) and kinking of the neural tube (*open arrowheads*). Scale bar, 500 μm. (TIFF)

**S3 Fig. *Tmem2^CKO^* embryos exhibit abnormal endocardial cushion formation and endocardial cell migration in the outflow tract region.** Transverse sections through the outflow tract region of E12.5 *Tmem2^CKO^* and control embryos were stained with anti-CD31 (*green*), anti-αSMA (*red*), and DAPI (*blue*). *Asterisk* indicates abnormal aggregates of endocardial cells in endocardial cushion mesenchyme. *Arrowheads* in the *Tmem2^CKO^* embryo point to lack of normal endocardial layer overlying the conotruncal endocardial cushions. Insets show enlarged images of the endocardial cushion mesenchyme. *Ao*, aorta; *ec*, endocardial cushion; *v*, ventricle. Scale bar, 200 μm. (TIFF)

**S4 Fig. Aberrant HA accumulation in the mandible and heart of E12.5 *Tmem2^CKO^* embryos.** Transverse sections of the mandible and the heart were stained with H&E or labeled with bHABP (*red*). Aberrant HA accumulation is observed in the mandible and the heart (areas indicated by rectangles are enlarged). *Asterisks* indicate endocardial cushion mesenchyme. Scale bar, 250 μm. (TIFF)

**S5 Fig. Generation of *Tmem2-FLAG^KI^* reporter mice.** (**A**) Schematic diagram of FLAG-tagged TMEM2 protein expressed from the *Tmem2-FLAG^KI^* locus. (**B**) Expression of TMEM2-FLAG protein in *Tmem2-FLAG^KI^* mice. TMEM2-FLAG protein was immunoprecipitated from the lysate of E11.0 whole embryos with anti-FLAG M2 antibody (*IP*). Precipitated materials were subjected to immunoblotting analysis (*IB*) with anti-TMEM2 antibody. (TIFF)

**S6 Fig. Expression of TMEM2 in the neuroepithelium.** Sagittal sections of *Tmem2-FLAG^KI^* reporter embryos at E11.0 were double-labeled with anti-FLAG (to detect TMEM2-FLAG protein) and anti-Nestin (to label neuroepithelial cells) antibodies. Areas indicated by boxes are enlarged in lower panels. *fb*, forebrain; *fp*, facial prominence; *mb*, midbrain. Scale bar, 50 μm. (TIFF)

**S7 Fig. Epithelial-mesenchymal transition and NCC specification are unaltered in Tmem2<sup>CKO</sup> mice.** (**A**) *In situ* hybridization in transverse sections shows that *Snail1* expression is detected in the dorsal neural tube (*brackets*) of control and *Tmem2<sup>CKO</sup>* embryos at E8.5. Scale bar, 250 μm. (**B**) Transverse sections of control and *Tmem2<sup>CKO</sup>* embryos at E8.0 were labeled with anti-Pax3 or anti-Pax7 antibody. Scale bars, 100 μm. *nt*, neural tube; *nf*, neural folds.
(TIFF)

**S8 Fig. Dorsoventral neural tube patterning in Tmem2<sup>CKO</sup> mice.** Transverse sections of the neural tube at the upper trunk level of control and *Tmem2<sup>CKO</sup>* embryos at E10.5 were labeled with anti-Nkx2.2 (a ventral marker) or anti-Pax7 (a dorsal marker) antibody. The overall patterning of the neural tube is not altered in *Tmem2<sup>CKO</sup>* mice. Scale bars, 500 μm.
(TIFF)

**S9 Fig. Generation of Tmem2-depleted O9-1 cells.** (**A**) Representative images of *Tmem2*-depleted and control O9-1 cells cultured on a regular culture dish (*left*). (**B**) Expression of *Tmem2* in these cells was evaluated by qPCR, with *Gapdh* as an internal control for normalization (*bar graph*). Means ± SD ($n = 5$) are shown as horizontal bars. ***$p < 0.001$ by unpaired Student's *t*-test. Scale bar, 5.0 μm.
(TIFF)

**S10 Fig. Integrin expression and differentiation capacity are not altered in O9-1 cells by Tmem2 deficiency. (A)** qPCR analyses of integrin α1 (*Itga1*), α2 (*Itga2*), β1 (*Itgb1*) and β3 (*Itgb3*). Data represent the mean ± SD (n = 3). *n.s.*, not significant by Student's *t*-test. (**B**) Images show chondrogenic (alcian blue staining) and osteogenic (alizarin-red staining) differentiation of control (*Control shRNA*) and *Tmem2*-depleted (*Tmem2 shRNA*) O9-1 cells. Scale bars, 200 μm.
(TIFF)

**S11 Fig. NCC migration on low molecular weight HA/Col1 substrate is not altered by Tmem2 deficiency.** (**A**) Representative images of migration of control (*Control shRNA*) and *Tmem2*-depleted (Tmem2 *shRNA*) O9-1 cells into a cell-free gap on low molecular weight HA/Col1 mixed substrates. Panels show images after a 24 or 48 h incubation. (**B**) Quantitative analysis of cell migration. Data represent the mean ± SD of the gap area covered by migratory cells relative to the area of the original gap (n = 3 per condition). *n.s.*, not significant by two-way ANOVA with Bonferroni's multiple comparison test. Scale bar, 200 μm.
(TIFF)

**S12 Fig. Analysis of apoptosis and proliferation of post-migratory NCCs in the branchial arches and the heart.** (**A**) Apoptotic cells were detected using TUNEL staining. *Tmem2<sup>CKO</sup>* embryos at E12.5 exhibit an increased number of TUNEL-positive cells in the branchial arches (*BA*). *Asterisk* indicates the outflow tract region. (**B**) Transverse sections of *Tmem2<sup>CKO</sup>* and control embryos at E12.5 were immunohistochemically stained with anti-PHH3 antibody and DAPI. Apoptosis is increased in the branchial arch of *Tmem2<sup>CKO</sup>* embryo compared to control embryos, whereas cell proliferation is not altered. Scale bar, 250 μm.
(TIFF)

**S13 Fig. Analysis of apoptosis in migrating NCCs and in early cranial NCC derivatives.** TUNEL staining (*red*) of the neural tube (**A**) and the maxillary component of the first brachial arch (**B**) of control and *Tmem2<sup>CKO</sup>* embryos at E9.0. NCCs are labeled by ZsGreen reporter

(*green*). *nt*, neural tube. Scale bar, 150 μm.
(TIFF)

**S1 Table. Primer sequences for PCR genotyping of mice.**
(DOCX)

**S2 Table. Sequences of crRNA and the inserted FLAG-encoding DNA fragment used for the creation of the *Tmem2-FLAG* knock-in allele (*Tmem2-FLAG^KI^*).**
(DOCX)

**S3 Table. Primer sequences for qPCR analysis of integrins.**
(DOCX)

## Acknowledgments

We thank Dr. Sachiko Iseki (Tokyo Medical and Dental University) for providing *Wnt1-Cre* mice. We thank Ms. Yuriko Nogami and Ms. Yuki Okamoto for the excellent care and maintenance of our mouse colony and for valuable assistance in the histological, molecular and protein work.

## Author Contributions

**Conceptualization:** Toshihiro Inubushi, Yu Yamaguchi.

**Data curation:** Toshihiro Inubushi, Makoto Abe, Hiroshi Kurosaka, Fumitoshi Irie.

**Formal analysis:** Toshihiro Inubushi, Yuichiro Nakanishi.

**Funding acquisition:** Toshihiro Inubushi, Takashi Yamashiro, Yu Yamaguchi.

**Investigation:** Toshihiro Inubushi, Yuichiro Nakanishi.

**Methodology:** Makoto Abe, Yoshifumi Takahata.

**Project administration:** Toshihiro Inubushi.

**Resources:** Yoshifumi Takahata, Riko Nishimura, Yu Yamaguchi.

**Supervision:** Takashi Yamashiro, Yu Yamaguchi.

**Validation:** Fumitoshi Irie, Yu Yamaguchi.

**Writing – original draft:** Toshihiro Inubushi.

**Writing – review & editing:** Toshihiro Inubushi, Fumitoshi Irie, Takashi Yamashiro, Yu Yamaguchi.

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
