## [Decision Letter · Decision Letter 0]

11 Oct 2021

Dear Dr Inubushi,

Thank you very much for submitting your Research Article entitled 'The cell surface hyaluronidase TMEM2 plays an essential role in mouse neural crest cell development and survival' to PLOS Genetics.

The manuscript was fully evaluated at the editorial level and by independent peer reviewers. The reviewers appreciated the attention to an important problem, but raised some substantial concerns about the current manuscript. Based on the reviews, we will not be able to accept this version of the manuscript, but we would be willing to review a much-revised version provided that the concerns of the reviewers can be addressed satisfactorily. We cannot, of course, promise publication at that time.

If you decide to revise the manuscript for further consideration at PLOS Genetics, please aim to resubmit within the next 60 days, unless it will take extra time to address the concerns of the reviewers, in which case we would appreciate an expected resubmission date by email to plosgenetics@plos.org.

[LINK]

We are sorry that we cannot be more positive about your manuscript at this stage. Please do not hesitate to contact us if you have any concerns or questions.

Yours sincerely,

Gregory Barsh

Editor-in-Chief

PLOS Genetics

Gregory Copenhaver

Editor-in-Chief

PLOS Genetics

Reviewer's Responses to Questions

**Comments to the Authors:**

Reviewer #1: Inubushi and co-workers present evidence that the hyaluronan-catabolizing protein transmembrane protein-2 (TMEM2) plays roles in the migration, differentiation and survival of neural crest cells. The manuscript is well written, and the study is well done. The data generally support the conclusions. The study will be of interest to the field.

One concern is that it is not cleare which cells express TMEM2. It would be helpful to use double-labeling assays to determine which other cells express tmem2. Are other neuroectodermal derivatives also positive?

The cell death studies are interesting but would benefit from a more detailed examination. Is there evidence that neural crest cells themselves are undergoing apoptosis in situ or only crest derivatives? Again, double labeling would be helpful

One question that arises from these findings is whether hyaluronan plays a role in neural crest cell differentiation. Although migration phenotypes are clear, how would loss of tmem2 influence differentiation if the crest cell line is challenged to differentiate into different neural crest lineages?

Work from Roberto Perris and others suggest that progenitor cell migration may require HA synthesis. The authors should therefore discuss the possibility that HA degradation products generated by Tmem2 or other HA-catabolizing proteins, as opposed to the loss of higher molecular weight HA, may signal neural crest cells to initiate emigration from the dorsal neural tube and/or contribute to other phenotypes identified in the CKO mice.

Reviewer #2: Inubushi et al. studied the functions of the cell surface hyaluronidase (TMEM2) in neural crest cell development.

Hyaluronan is a very important extracellular matrix. In this study, the authors induced excess hyaluronan presence by depleting one of hyaluronidases, Tmem2.

Tmem2 is definitely required for NCC development. And it is involved in NDD emigration, cell death. The authors introduce the mutant dies with craniofacial and cardiovascular abnormalities. Craniofacial abnormalities would be explained by dcreased number of NCCs and cell death. But how do the authors explain about cardiovascular abnormalities?

Since TMEM2 is ubiquitously expressed and consistently required, neural crest cells are constantly affected by excess hyaluronan in the mutant. After NCC migration, excess hyaluronan must affect the space for the cells, therefore, reduced number of neural crest cells in the trigeminal ganglion is reasonable. If there is not enough physical space, the number of the cells should be reduced. Do the dorasal ganglia show cell death? If not, why is the cell death observed in craniofacial mesenchyme?

In neural crest development, after the commitment there is a step of epithelial-mesenchymal transition, then the cells delaminate. The word delamination might include the step of epithelial-mesenchymal transition, however, epithelial-mesenchymal transition is supposed to be an important step for neural crest appearance in general. Therefore, neural crest development process should be carefully explained. In addition, in the mutant does epithelial-mesenchymal transition of NCCs complete?

It is also important to describe the definition of embryonic development stage. When is the day 0?

The authors observed the domain occupied by Z/EG positive cells within the dorsal neural tube at E10.5. Neural crest cells arise at the border of neural plate and the surface ectoderm. Does Sox10 expression pattern in tne mutant show the NCCs migrate within the neural tube towards ventral side?

In addition, the role of Sox9 and Sox10 expression in NCCs should be clearly defined.

What is the rationale to use the substrate for OP9 cells to mimic the environment of emigration of NCCs?

In Fig. 1, Trem2 is expressed in the dorsal part of the neural tube. In mice cranial neural crest cells appear and start migration before neural tube closure. In this image it is not clear which level of the transverse section is shown.

The authors report that NCC express Tmem2 and there are

In some of the images, the lettering is too big to understand the phenotype.

In Fig. 1C, legend says “Mild exencephaly-like defects”. Since the authors observed the fetus, it should be clearly mentioned. Or does “exencephaly-like” indicate another condition?

Reviewer #3: While previous works have demonstrated the critical nature of HA production for neural crest cell (NCC) migration and the development of NCC-derived structures, this paper contributes to the lesser-known role of HA breakdown by hyaluronidases in neural crest development. The authors show that Sox9+ emigrating cells are TMEM2+, and TMEM2 mutants have NCC migration defects. Consequently, mouse embryos with an NCC-specific knockout of TMEM2 have malformations of craniofacial and cardiovascular structures that lead to embryonic lethality before E13.5. Using in vitro methods, the authors postulate that TMEM2 is needed to degrade HA locally in order to form focal adhesions to the surrounding matrix so cells can migrate. This work is presented with compelling imaging of mouse tissues and in vitro assays, but overstates that TMEM2 contributes to migration/morphogenesis of the neural crest – at most it appears that TMEM2 is an important regulator of neural crest cell survival. What would be more compelling and novel is to answer why and by what mechanisms ectopic maintenance of HA leads to cell death. It is also unclear when cell death occurs and whether presumptive migration defects are not secondary to cell death. This paper contributes to our growing knowledge of TMEM2 and its role in the embryo but falls short on providing mechanists understanding of what TMEM2 is doing in the crest – no direct evidence for migration defect in light of apoptosis. Lastly, the authors need to better connect to existing literature about the relationship between HA and NCC migration/development.

Major:

1. The connection to cell emigration defect is not well supported especially in light of very significant apoptosis of the mutant crest cells. It is unclear when apoptosis initiates, the data are shown only for E12.5 but how about earlier? If there is earlier apoptosis it would explain why there are fewer Sox positive cells (cell death/survival versus migration defects – less cells surviving hence less cells will emigrate, time kinetics of cell death will be crucial in this regards). it will also be important to show with double-staining that the apoptotic cells are Sox9 and Sox 10 positive. Do you see cell death in vitro and can this explain the inability to create a wound in your would healing assay.

2. The authors don't examine the possibility that neural tube patterning may be affected in the mutants. It would be important to provide gene expression analysis of appropriate markers. Patterning defects could be an underlying reason for some of the later phenotypes especially in Figure 4B demonstrating misfolded midbrain.

3. It is interesting that the hearts do not show apoptosis - it would be important to show whether the localization of vinculin remains affected. If no, it is feasible that vinculin defects in the crest are secondary to apoptosis – this is crucial aspect to address – what comes first, survival defect then focal adhesion lesions or vice versa. If hearts have defects in vinculin localization without apoptosis, this could serve as an important control for the former. In this regard, would be powerful to compare these phenotypes to well-characterized crest cell migration defect models.

4. If the major phenotype of mutant cells is apoptosis, it would be very interesting to investigate why loss of TMEM, and ectopic/abnormal maintenance of HA induces apoptosis and whether this is a specific feature of neural crest versus cardiac crest cells – these data would contribute novel insights to our current understanding. Similarly, if the authors claim migration as the key defect (no evidence of cell death in emigrating cells or earlier), collective polarized migration is well described for the crest cell and should be investigated in vivo with appropriate markers and imaging. Just showing less Sox+ cells is not sufficient to claim emigration defects.

5. The paper would benefit from a more thorough description of existing literature about HA and NCC migration. While mentioned briefly in the introduction, the reader may not fully appreciate that HA production, as well as catabolism as shown in this paper, is absolutely critical for proper NCC migration and the development of NCC-derived structures. Related to this, the authors say that HA is a barrier against NCC migration in vitro (line 390). However, it is my understanding that HA is pro-migratory for NCCs in vivo—Has1 and Has2 are both needed for proper NCC migration in Xenopus embryos (Casini, Nardi, and Ori, 2012), while versican acts as a barrier against NCC migration (Landolt et al., 1995). While the data in figure 5 supports your argument, I think it is important to give the full picture of the in vivo data that we have about NCC migration and the ECM. Similarly, when Has2 is conditionally knocked out in cranial NCCs, mutant mice have cleft palate due to diminished palatal shelves that don’t move and fuse properly (Yonemitsu, Lin, and Yu 2020). What explanation do you propose for both a hyaluronidase (TMEM2 and some Hyal2 null—Muggenthaler et al., 2017) and an HA synthase having the same phenotype when knocked out? Does the importance of both HA production and HA degradation suggest that it is in fact HA fragments that are key for NCC migration? This could be tested in your in vitro system, where you could make separate Col1/HA substrates with high molecular weight HA or with HA fragments, and see if the migration defect exhibited by Tmem2 shRNA cells is rescued when grown on substrate with HA fragments.

Minor:

1. Line 29: “regulated in during development”—change to “regulated during development”

2. Line 60: change NSC to NCC

3. Lines 87-90—The authors try to connect the Has2 KO phenotypes in cardiac cushions with cranial NCCs, but with very weak evidence

4. Figure 1b: would be better to have an image of a mutant embryo that doesn’t have tissue damage (forelimb looks like it was almost cut off, at first glance this appears to be a phenotype)

5. Figure 1f: could quantify the cell-free spaces in the H&E images to further illustrate the difference in ECM between wild-type and mutant. Similar could possibly be done for Figure S3.

6. Line 395: “stems” instead of “steams”

7. Have you checked if integrins are present at the same number on TMEM2 knockdown cells as on wild-type cells? If integrins were downregulated that would be an alternate explanation for lack of focal adhesions (unlikely, but good to check).

8. In vitro migration assay, can you rescue with transfection of TMEM2

9. Line 309: Sox10 staining (Fig 3B) demonstrates that the number of NCCs migrating along the ventral migration pathway is reduced”. What about a pan marker of crest, as possible that total number is reduced so migrating number will also be reduced.

10. Line 336 spelling: lineage tracing not trancing

11. Line 339: briefly describe what this strain is: ZsGreen (Z/EG)

12. Line 404: what is the evidence that HA is being degraded here

13. Is HA accumulation also seen in the mutant hearts or neural tube specific?

14. Time kinetics of TMEM expression prior to E9, when does expression initiate? Does it coincide with NCC specification or migration?

**Have all data underlying the figures and results presented in the manuscript been provided?**

Reviewer #1: Yes

Reviewer #2: Yes

Reviewer #3: Yes

PLOS authors have the option to publish the peer review history of their article (what does this mean?). If published, this will include your full peer review and any attached files.

Reviewer #1: No

Reviewer #2: No

Reviewer #3: No

---

## [Decision Letter · Decision Letter 1]

21 Mar 2022

Dear Dr Inubushi,

Thank you very much for submitting your Research Article entitled 'The cell surface hyaluronidase TMEM2 plays an essential role in mouse neural crest cell development and survival' to PLOS Genetics.

The revised manuscript was seen by the original two reviewers. As you will see, the reviewers (and the editors) agree that the revision is improved; however, there are still some key issues that would need to be addressed in order to move forward, and that will require additional experiments and data.

Based on the reviews and our editorial evaluation, we will not be able to accept this version of the manuscript, but we would be willing to review a much-revised version. We cannot, of course, promise publication at that time.

If you decide to revise the manuscript for further consideration at PLOS Genetics, please aim to resubmit within the next 60 days, unless it will take extra time to address the concerns of the reviewers, in which case we would appreciate an expected resubmission date by email to plosgenetics@plos.org.

[LINK]

We are sorry that we cannot be more positive about your manuscript at this stage. Please do not hesitate to contact us if you have any concerns or questions.

Yours sincerely,

Gregory Barsh

Editor-in-Chief

PLOS Genetics

Gregory Copenhaver

Editor-in-Chief

PLOS Genetics

Reviewer's Responses to Questions

**Comments to the Authors:**

Reviewer #2: The manuscript entitled “The cell surface hyaluronidase TMEM2 plays an essential role in mouse neural crest cell development and survival Short title: Role of TMEM2 in neural crest development” has been revised.

The manuscript shows the importance of TMEM2 in craniofacial development. The main points of the authors are TMEM2 is required for cranial neural crest cell migration and survival of the cells after they arrive in their destinations. The way how the authors present the results is too rough. For instance, the authors write “Fig. 3C graph shows the quantification of migrating NCCs in Tmem2CKO;Z/EG and controlembryos at E10.5.”, however, neural crest cell migration completes by E10.5. Further, some of the results needs higher magnification images to understand what the authors write in the text. The authors are advised to consider which data sets clearly indicate what they want to inform to the readers.

Depletion Tmem2 in OP-9 induced cell death, but not all the OP-9 cells. Large portion of craniofacial mesenchyme is neural crest origin, however, the region of cell death is limited. Is it possible to explain?

Line 204, the word “hereafter” is duplicated.

Line361-362, what does the phrase “whereas the domain occupied by Z/EG362

positive cells within the dorsal neural tube is expanded in Tmem2CKO;Z/EG embryos” mean?

Reviewer #3: This reviewer appreciates the detailed responses; however, the following points (5 and 12) need quantifications, one cannot conclude significant changes without statistics.

Along the same lines, point 9 is important if the authors want to conclude that the number of Sox10 cells is reduced, if this is significantly reduced is has to be normalized to the total number of cells.

Finally, in the new figure (S8) (point 2) - it is important to specify whether this is cranial/caudal neural tube, since in the other figures (Fig. 2 and 3), the caudal neural tube always looks fine and it’s only the cranial neural tube where phenotypes are seen.

Points 5, 9 and 12:

5. Figure 1f: could quantify the cell-free spaces in the H&E images to further illustrate the difference in ECM between wild-type and mutant. Similar could possibly be done for Figure S3.

We believe that the significant increase in extracellular spaces in mutant mice is obvious without quantification. Also, since H&E staining does not define cell contours (hematoxylin stains nuclei), it is not optimal for the quantitative analysis of the size of extracellular space. Accordingly, we chose not to perform the quantification.

9. Line 309: Sox10 staining (Fig 3B) demonstrates that the number of NCCs migrating along the ventral migration pathway is reduced”. What about a pan marker of crest, as possible that total number is reduced so migrating number will also be reduced.

Due to time constraint, we did not perform this analysis. However, we believe that our data as a whole, including those with Wnt1-Cre–dependent ZsGreen-labeling of NCCs, have firmly established the role of TMEM2 in NCC migration, even without this requested analysis.

12. Line 404: what is the evidence that HA is being degraded here

Dark holes/streaks in the green HA substrates represent the sites where HA is degraded by cells. Please refer to Yamamoto et al., J. Biol. Chem. 292:7304-7313) for the basis of the assay.

**Have all data underlying the figures and results presented in the manuscript been provided?**

Reviewer #2: Yes

Reviewer #3: Yes

PLOS authors have the option to publish the peer review history of their article (what does this mean?). If published, this will include your full peer review and any attached files.

Reviewer #2: No

Reviewer #3: No

---

## [Decision Letter · Decision Letter 2]

29 Jun 2022

Dear Dr Inubushi,

We are pleased to inform you that your manuscript entitled "The cell surface hyaluronidase TMEM2 plays an essential role in mouse neural crest cell development and survival" has been editorially accepted for publication in PLOS Genetics. Congratulations!

The revised manuscript was seen by two of the previous reviewers. As you will see, both reviewers are enthusiastic. However, reviewer #2 points out that there is an additional recent relevant publication. We ask you to incorporate a citation to this publication in your manuscript which can be done during the production process.

Yours sincerely,

Gregory S. Barsh

Editor-in-Chief

PLOS Genetics

Gregory Copenhaver

Editor-in-Chief

PLOS Genetics

Comments from the reviewers (if applicable):

Reviewer's Responses to Questions

**Comments to the Authors:**

Reviewer #2: The manuscript has been revised according to the comments.

I just came across the new manuscript by Shellard and Mayor on neural crest emigration, Collective durotaxis along a self-generated stiffness gradient in vivo, https://doi.org/10.1038/s41586-021-04210-x. The authors quoted the works from them in the text. I am wondering if the authors could replace one of them with the very recent one.

Reviewer #3: The authors have addressed all of my concerns.

**Have all data underlying the figures and results presented in the manuscript been provided?**

Reviewer #2: Yes

Reviewer #3: Yes

PLOS authors have the option to publish the peer review history of their article (what does this mean?). If published, this will include your full peer review and any attached files.

Reviewer #2: No

Reviewer #3: No

**Data Deposition**

http://datadryad.org/submit?journalID=pgenetics&manu=PGENETICS-D-21-01060R2

**Press Queries**

---

## [Editor Report · Acceptance letter]

12 Jul 2022

PGENETICS-D-21-01060R2 

The cell surface hyaluronidase TMEM2 plays an essential role in mouse neural crest cell development and survival 

Dear Dr Inubushi, 

We are pleased to inform you that your manuscript entitled "The cell surface hyaluronidase TMEM2 plays an essential role in mouse neural crest cell development and survival" has been formally accepted for publication in PLOS Genetics! Your manuscript is now with our production department and you will be notified of the publication date in due course.

With kind regards,

Olena Szabo

PLOS Genetics

On behalf of:
